mathematical physics/biomechanics/
mathematical modelling

haemodynamics, arteriovenous malformation,
optimal control, two-phase filtration,
CABARET scheme

**Author for correspondence:**
Alexandr A. Cherevko
e-mail: cherevko@mail.ru

Electronic supplementary material is available
online at https://doi.org/10.6084/m9.figshare.c.
5065281.

# Modelling of the arteriovenous malformation embolization optimal scenario

Alexandr A. Cherevko[1,2], Tatiana S. Gologush[1],
Irina A. Petrenko[3], Vladimir V. Ostapenko[1,2]
and Vyacheslav A. Panarin[4]

[1]Lavrentyev Institute of Hydrodynamics of the Siberian Branch of the Russian Academy of Sciences, 630090 Novosibirsk, Russia
[2]Hydrodynamics Department, Novosibirsk State University, 630090 Novosibirsk, Russia
[3]Functional Analysis and its Applications Department, Vladimir State University, 600000 Vladimir, Russia
[4]Medical Center of Far Eastern Federal University, 690090 Vladivostok, Russia

AAC, 0000-0003-2014-854X

Cerebral arteriovenous malformation (AVM) is a congenital brain vessels pathology, in which the arterial and venous blood channels are connected by tangles of abnormal blood vessels. It is a dangerous disease that affects brain functioning causing the high risk of intracerebral haemorrhage. One of AVM treatment methods is embolization—the endovascular filling of the AVM vessel bundle with a special embolic agent. This method is widely used, but still in some cases is accompanied by intraoperative AVM vessels rupture. In this paper, the optimal scenario of AVM embolization is studied from the safety and effectiveness of the procedure point of view. The co-movement of blood and embolic agent in the AVM body is modelled on the basis of a one-dimensional two-phase filtration model. Optimal control problem with phase constraints arising from medicine is formulated and numerically solved. In numerical analysis, the monotone modification of the CABARET scheme is used. Optimal embolization model is constructed on the basis of real patients' clinical data collected during neurosurgical operations. For the special case of embolic agent, input admissible and optimal embolization scenarios were calculated.

# 1. Introduction

Cerebral arteriovenous malformation (AVM) is a congenital abnormality of brain vessels, in which a direct discharge of blood

from the arterial blood pool into the venous pool is present, bypassing the capillary vessel network [1–5]. Due to capillary network absence, the resistance of the corresponding part of circulatory system decreases. These deviations in vascular system modify both haemodynamic parameters (flow velocity and pressure) and strength properties of blood vessels. Most often, AVMs become symptomatic at the age of 20–40 years and are detected during this period of life. Moreover, in more than half of cases the disease manifests itself by haemorrhage. Disability related to haemorrhage from AVM reaches 50% with a risk of death of 10–29%. For AVM posterior cranial fossa, the mortality rate exceeds 50%.

The haemodynamic associated with intracranial AVMs is complex and varies as morphology and angioarchitecture changes, especially with different treatment methods. Various treatment methods effects on AVM and its environment haemodynamics are presented, for example, in the review of [6].

At the moment, the most common methods of AVM treatment are microsurgical, endovascular and radiosurgical. Endovascular embolization is the selective deactivation of blood vessels from the bloodstream by filling them with a special embolic agent. Modern level of medicine development makes embolization the most preferable method due to minimal invasiveness and possibility of operating in the deep, functionally significant brain areas [7]. This method is widely used. Despite the well-developed embolization technique, the risk of intraoperative vascular rupture remains a concern. According to a study of 408 patients with AVM who were treated endovascularly, it was noted 11% frequency of haemorrhagic complications associated with the treatment [8]. Thus, modelling of AVM embolization is of current interest.

By type of vessels, connecting the arterial and venous basins, AVMs can be divided into fistula (consisting of large vessels) and racemic (consisting of large number of small diameter vessels, which are chaotically intertwined and intersect each other). There are also mixed-type AVMs that combine both racemic and fistula parts. Modern medical examination methods such as CT, MRI and cerebral angiography do not allow to determine the detailed geometric AVM structure *in vivo*, since its nidus (tangles of abnormal blood vessels) consists of very large number of thin vessels with diameter down to 0.1 mm which exceeds medical equipment resolution. Therefore, various simplified methods for AVM modelling are needed.

Mathematical description of blood flow is often based on one-dimensional approximation of Navier–Stokes equations by averaging over blood vessel cross-section. Analytical studies of equations describing flows in pipes with rigid and elastic boundaries can be found in [9–11]. Viscous fluid flow in a network of soft tubes is modelled in [12–14] on the base of one-dimensional approximation of mass and momentum conservation laws. Numerical simulation of haemodynamics for large blood vessels based on three- to one-dimensional coupled flow is considered, for example, in [15–17].

AVM and blood flow in surrounding vessels flow interaction is often studied in analogy with electric and hydraulic networks [18–20]. Such models make it possible to evaluate the effect of various embolization scenarios on blood flow restructuration and are consistent with the general medical view on AVM haemodynamics. The effect of AVM on the blood flow in the surrounding vessels has also been studied using the mass and momentum conservation laws of the ideal incompressible fluid flow. The resulting equations form a hyperbolic system of differential conservation laws. One-dimensional variant of this system on a vessel graph was studied in [21,22], and its two-dimensional variant on branching network channels was considered in [23,24]. The process of embolization was studied in [25] where two-phase flow model was used for description of the viscous fluid drop movement through a bifurcation point. In [26], it was studied a two-phase model of AVM embolization where embolic agent and blood interaction along with its hardening is simulated by viscosity increase.

The above literature review shows the great interest of the scientific community in this topic. At the same time, the diversity of approaches and models indicates the presence of a fairly large number of open questions. This is due to the fact that an exhaustive anatomical and physiological description of AVM requires detailed knowledge of malformation nidus structure and mechanical and physiological properties of vessels forming it. Modern diagnostic equipment arsenal does not provide all the necessary information. In this regard, used models have sufficiently strong assumptions and simplifications, which, however, allow to obtain qualitatively correct description of haemodynamics in studied anomaly.

In this paper, the flow of blood and embolic agent through the AVM body was considered as filtration flow on the basis of Darcy's Law. Such an approach, in our opinion, is justified for small-vascular racemic AVM compartments' embolization description. Some assumptions on embolic agent behaviour, spatial characteristics of AVM, blood rheology and wall properties are made. Specifically, both liquids are assumed to be Newtonian with constant viscosities, incompressible and immiscible. In numerical calculations, real blood and embolic agent (ONYX18) viscosities were used. AVM is simplified to one-dimensional case with even distribution of its physical characteristics (porosity, permeability and

cross-sectional area) along model length. Some aspects such as capillary forces, embolic agent adsorption effect and wall properties are neglected.

To investigate this process, one-dimensional two-phase filtration model is used leading to the special Buckley–Leverett equation in §2. This mathematical model represents a scalar conservation law with non-convex flux for the numerical solution of which we will use a monotonic modification of the second order CABARET scheme [27], see §3. This scheme demonstrates higher accuracy in the localization of strong and weak discontinuities compared to other classes of high-order shock capturing schemes [28–32] for which monotonization various nonlinear flux correction procedures are used.

Initial-boundary value AVM problem is considered in §4. Numerical modelling of the problem under consideration allows us to describe the evolution of the embolization front, taking into account its partial decay which leads to residual blood content in embolized AVM part, see §6. Cardiac cycle effect could be ignored due to modified time used in the problem. Reverse transition to the physical time allows to build a solution corresponding to any given cardiac flow–time dependency. Optimal embolization problem is stated based on this model. Admissible and optimal embolization scenarios, which reduce the risk of rupture of AVM vessels, are considered in §7. Mathematically, it is optimal control problem with integral objective function for partial differential equation with state constraints considered in §§5 and 6. Time-dependent boundary condition function is used as control, while constraints arise from medical aspects. The proposed model takes into account the peculiarities of actual embolization process. AVM nidus absolute and relative permeability functions are constructed using intraoperative intracranial measurements in appendices A and B. Mathematical model based on intraoperative data from real patients allowed us to create patient-specific models. This approach can potentially be used in neurosurgical practice to provide a neurosurgeon with recommendation on the course of operation.

## 2. Model description

Racemic AVM compartments with sufficient modelling accuracy can be considered as a porous medium and are the subject of present research. Blood and embolic agent movement through racemic AVM can be modelled as two-phase filtration process, where the displaced phase is blood, and displacing phase is embolic agent. Blood and embolic agent are considered as incompressible and immiscible liquids. We neglect the interfacial capillary forces—in this case the pressure in the phases is assumed to be the same. In the one-dimensional case, the displacement of one phase by another allows a mathematical description first proposed by Buckley & Leverett [33]. This description is based on the notions of saturation, absolute and relative phase permeabilities [34]. We denote the local AVM blood saturation during embolization (blood concentration) as $S(x, t) \in [0, 1]$. Now embolic agent concentration is $1 - S(t, x)$. Then Buckley–Leverett equation for blood concentration $S(t, x)$ looks as follows:

$$m S_\theta + Q/A f(S)_x = 0, \tag{2.1}$$

where $\theta$ is physical time, $x$ is the spatial coordinate, $m$ is AVM nidus porosity, $A$ is AVM cross-section, $Q$ is volumetric flow rate of two-phase mixture. The saturation function $f(S)$ is the Buckley–Leverett function and is equal to the volume fraction of the displaced liquid (blood) flow in the total flux of the two phases.

We assume that the structure of the AVM is homogeneous. Further the side surface of the AVM is considered impermeable, the cross-section, porosity and other parameters of the AVM are assumed to be constant in length and in time. Under these assumptions, the equation for $S(t, x)$ could be written in simplified form [34]

$$S_t + f(S)_x = 0, \tag{2.2}$$

where $t$ is modified length dimension time

$$t = \frac{1}{m A} \int Q(\theta) d\theta, \quad t(0) = 0, \tag{2.3}$$

where $\theta$ is physical time, $m$ is AVM nidus porosity, $A$ is AVM cross-section, $Q(\theta)$ is volumetric flow rate of two-phase mixture. Note that the function $Q$ does not depend on $x$ because of AVM side surface impermeability and phases incompressibility assumptions. In the modified time $t$, the total volumetric flow of two phases and porosity can be considered constant and equal to 1. Various cardiac cycle types could be considered using different forms of function $Q(\theta)$. Inverse time modification leads to the solution of equation (2.1) corresponding to physical time–flow relation.

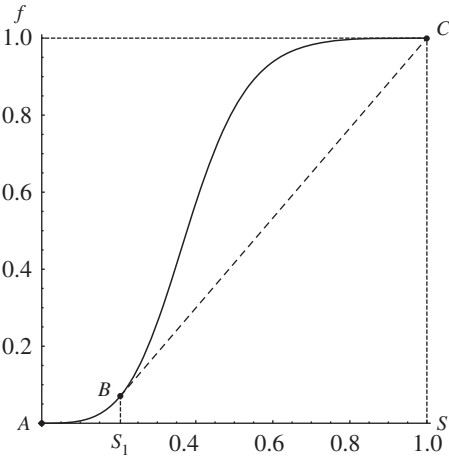

**Figure 1.** Typical view of Buckley–Leverett function (2.4).

The Buckley–Leverett function $f(S)$ mathematically could be presented by formula

$$f(S) = \frac{Q_b}{Q} = \frac{k_b(S)/\eta_b}{k_b(S)/\eta_b + k_e(S)/\eta_e}. \tag{2.4}$$

Here $Q_b$ is blood volume flow; $k_b$, $k_e$ are phase relative permeabilities and $\eta_b$, $\eta_e$ are phase dynamic viscosity coefficients, where index $b$ corresponds to blood and index $e$ to embolic agent. Function $f(S)$ increases monotonously from 0 to 1 as blood concentration $S$ grows (figure 1). This function always has an inflection point, which separates concavity and convexity segments. Therefore, the Buckley–Leverett model (2.2) and (2.4) represents the scalar conservation law with a non-convex flux, which allows increasing and decreasing shock waves and also rarefaction waves, as well as composite shock–rarefaction waves. The small perturbations in this model propagate with speed $f'(S) \geq 0$, which depends on $S$ in a non-monotonic way and in this paper additionally satisfies the often used conditions,

$$f'(S) > 0 \quad \forall S \in (0, 1), \quad f'(0) = f'(1) = 0. \tag{2.5}$$

# 3. Numerical method

In [35], the second-order Upwind Leapfrog scheme was proposed for hyperbolic equations' numerical solution. A detailed analysis of this scheme was carried out in [36,37], where taking into account the skew-symmetry of its stencil (which is two-point in space and three-layer in time) it was called the CABARET scheme. The main advantages of the CABARET scheme are determined by its compact spatial stencil and for linear transport equation scheme time reversibility and approximation accuracy with two different Courant numbers $r = 0.5$ and $r = 1$, which gives it unique dissipative and dispersive properties [37]. For gas dynamics equations numerical simulation, a balance-characteristic version of CABARET scheme was developed by Goloviznin [38]. Taking into account flux variables correction, this scheme showed high accuracy in Blast Wave test calculation [39].

Currently, for numerical simulations of spatially multidimensional gas-dynamic [40] and hydraulic [41] flows, the two-layer in time form of the CABARET scheme [42] is widely used. The monotonicity of this scheme for approximation of scalar conservation law with convex flux was studied in [43] and with a non-convex flux in [27]. In the present paper, for Buckley–Leverett equations (2.2) and (2.4) numerical solution the monotonic modification of the CABARET scheme proposed in [27,44] will be used.

# 4. Initial-boundary value arteriovenous malformation problem

We denote the AVM length by $L$ and total embolization time by $T$. Initially the AVM is completely filled with blood and does not contain embolic agent. Therefore, the initial data for the Buckley–Leverett equation (2.2) with flux function (2.4) is

$$S(0, x) = S_0(x) \equiv 1, \quad x \in [0, L]. \tag{4.1}$$

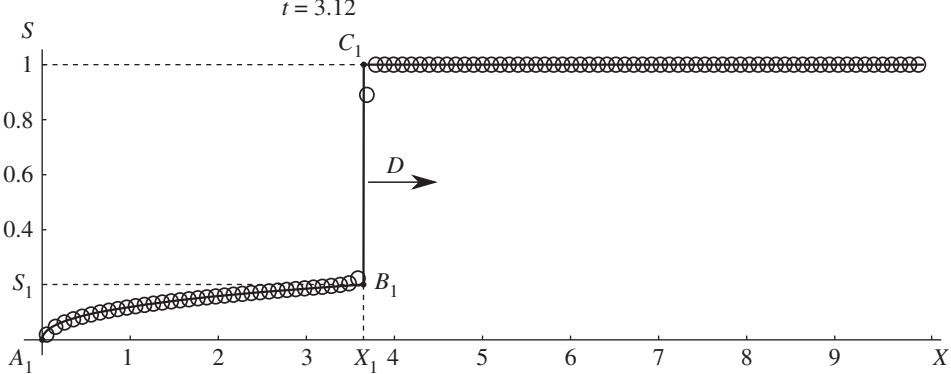

**Figure 2.** Exact (line) and numerical solution (circles) comparison for Buckley–Leverett function $f(S) = (S^3/4)/((1 - S)^3/9 + S^3/4)$ with $t < t_1$.

The only input of the embolic agent and blood mixture to the AVM is made through the feeding artery, which is set to be the origin $x = 0$. Blood concentration here is given by the function $g(t)$, determined by the chosen embolization strategy. Thus we obtain the following boundary condition:

$$S(t, 0) = g(t) \in [0, 1], \quad t \in (0, T]. \tag{4.2}$$

As a result, equations (2.2), (2.4), (4.1) and (4.2) set the desired initial-boundary value AVM problem. It is correct because the characteristics of equation (2.2) propagate in the positive direction of the $x$-axis while $f'(S) > 0$, which is true for $S \in (0, 1)$.

Experience of some neurosurgical operations shows that natural course of the operation is total AVM cross-section overlap by embolic agent (which mathematically is $g = 0$) followed by embolic agent concentration reduction (blood concentration increase) in AVM input. Thus, the boundary function $g(t)$ could be chosen as follows:

$$g(t) = \begin{cases} 0, & 0 < t < t_1, \\ g^*(t), & t_1 \leq t < t_2, \\ 1, & t_2 \leq t \leq T, \end{cases} \tag{4.3}$$

where $g^*(t)$ is smooth strictly monotonically increasing function satisfying conditions $g^*(t_1) = 0$ and $g^*(t_2) = 1$.

From the theory of a weak solution of the hyperbolic conservation laws with a non-convex flux [45], it follows that in the initial time interval $[0, t_1)$ the exact solution of the problem under consideration is an increase shock (vertical segment $B_1 C_1$ in figure 2), to which the centred rarefaction wave (line $A_1 B_1$ in figure 2) is adjacent to the left. This discontinuous solution is given by the formula

$$S(x, t) = \begin{cases} a^{-1}(x/t), & 0 \leq x < Dt, \\ 1, & Dt < x \leq L, \end{cases} \tag{4.4}$$

where $a^{-1}$ is function, inverse to function $f'$;

$$D = \frac{1 - f(S_1)}{1 - S_1} \tag{4.5}$$

is the shock velocity, which is determined from the Hugoniot condition for the conservation law (2.2); $S_1$ is the horizontal coordinate of the point $B$ in figure 1 in which the straight line passing through the point $C = (1, 1)$ touches the graph of the function $f(S)$; the value of $S_1$ is calculated from the equation

$$f(S_1) + f'(S_1)(1 - S_1) = 1. \tag{4.6}$$

Solid line in figure 2 shows exact solution (4.4) and (4.5) for Buckley–Leverett function $f(S) = (S^3/4)/((1 - S)^3/9 + S^3/4)$ at time $t = 3.12 < t_1$ while circles present numerical solution obtained by CABARET scheme. One can see a good concurrence between numerical and exact solutions both at discontinuity and in rarefaction wave region. Such comparison of exact and numerical solutions for $t < t_1$ was used to control the accuracy of calculations with various clinically based Buckley–Leverett functions (appendix B).

On the time interval $[t_1, T]$, the exact solution of problems (2.2), (2.4), (4.1) and (4.2) becomes more complex and for its calculation we will use the CABARET scheme. A significant advantage of the CABARET scheme compared to WENO type schemes [27] is that this scheme is defined on a compact three-point spatial stencil located inside a single cell of the difference grid, and therefore, when calculating initial-boundary value problems, it is not necessary for the CABARET scheme to apply auxiliary asymmetric difference equations in the near-boundary grid nodes.

To solve optimal embolization problem, Buckley–Leverett functions (2.4) were constructed for several patients using clinical data of haemodynamic parameters from neurosurgical operations at National Medical Research Center named after academic E. N. Meshalkin [46]. Clinically based Buckley–Leverett function construction is given in appendix B.

# 5. Unit load as restriction factor

The choice of the value characterizing the risk of AVM rupture is an important question arising in the study of embolization process. To answer this question, the notion of unit load was introduced [47]. Unit load is the mean value of energy released per unit of time with blood passing through the unit of AVM volume. It is given by formula $(E_{in} - E_{out})/V$, with $E_i = Q_i(t)P_i(t)$, where $E_i$ are blood energy flows on AVM nidus input and output; $Q_i$, $P_i$ are volumetric blood flow and total blood pressure with $i \in$ (in, out). $V(t)$ is the AVM volume part occupied by blood.

AVM embolization degree $D(t)$ is another important parameter and is defined as the fraction of AVM vascular space occupied by embolic agent. It can vary from 0 to 1 and for the considered embolization model can be calculated by formula

$$D(t) = 1 - \frac{1}{L} \int_0^L S(t, x) \, \mathrm{d}x. \tag{5.1}$$

Unit load behaviour for real patients using clinical data was studied in [47]. The connection between unit load and AVM embolization degree was established. It was noted that unit load begins to increase sharply with embolization degree of more than 60%. Two patient groups were studied in National Medical Research Center named after academic E. N. Meshalkin. In the first group, the unit load sharp increase near 60% embolization degree was taken into account and operations ended earlier than this value was achieved. In the second group, the operations were carried out without taking into account the unit load. It was shown that accounting for the unit load reduced the risk of complication (haemorrhage). It should be noted that embolization degree can differ from 60% and is determined individually by neurosurgeon for each patient, based on the patient's condition. For example, for small malformations with one supply vessel, a complete embolization is usually performed in one stage. Further in the paper without loss of generality, the case of the 60% maximum admissible embolization degree is considered.

It is known that simultaneous exclusion of high flow AVMs is accompanied by elevated risk of unfavourable bloodstream redistribution and, as a result, perioperative haemorrhage. In this regard, there is a practice of embolization dividing into stages. In current study, embolization degree is used as the value characterizing the risk of AVM rupture. On the one hand, value of embolization degree greater than 60% is considered dangerous. On the other hand, too small values of embolization degree show that operation was not effective since the pathological vascular bed remains widely open to blood flow and the number of surgical interventions unjustifiably increase. Thus, it is necessary to find some compromise between the risk of complications and the most complete exclusion of the AVM from the bloodstream per one stage.

# 6. Blood conductivity as embolization success factor

Along with embolization degree which characterizes AVM fullness by embolic agent, its distribution inside AVM is important. Residual blood flow through the AVM is determined by how much we can reduce AVM blood conductivity, which directly depends on embolic agent distribution in AVM nidus. Blood conductivity can be determined by Darcy's differential law for blood,

$$Q_b = -KA \frac{k_b(S)}{\eta_b} \frac{\partial p}{\partial x}. \tag{6.1}$$

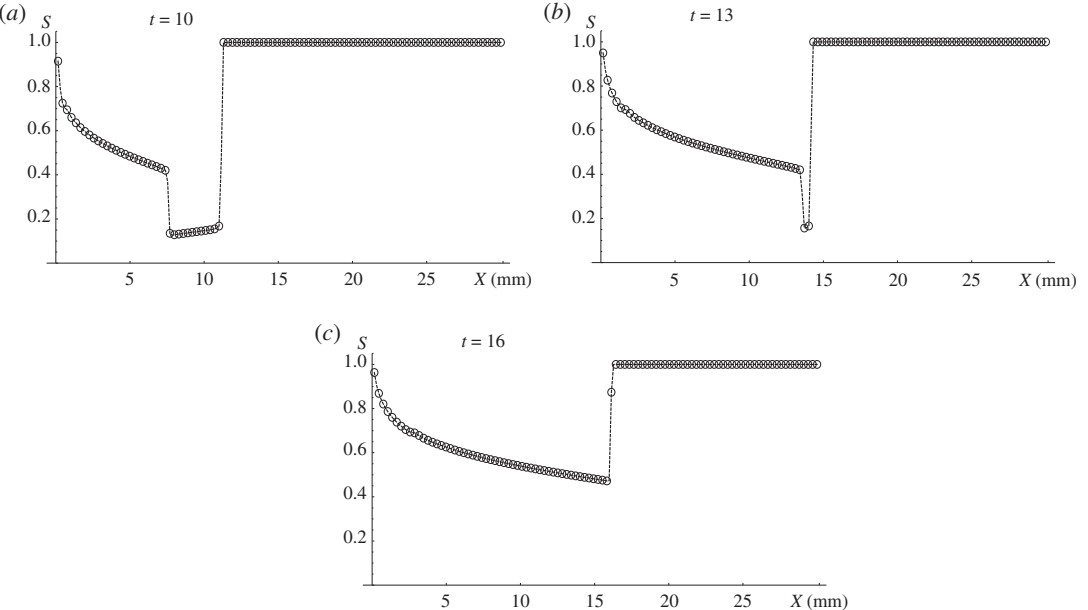

**Figure 3.** Bad embolization scenario for patient P. Consecutive stages of the process development at (a) $t = 10$, (b) $t = 13$, (c) $t = 16$. Black dots stand for numerical result and the dashed line is its linear interpolation.

Here $p(t, x)$ is pressure distribution and $Q_b(t, x)$ is blood volume flow distribution in the AVM. Thus, AVM local unit resistance to advancement of a blood flow is $R(t, x) = \eta_b / (K A k_b(S))$. Then AVM full blood conductivity is given by formula

$$\hat{G}(t) = \left( \int_0^L R(t, x)dx \right)^{-1}. \tag{6.2}$$

To determine the residual blood flow, it is convenient to use dimensionless quantities. Let us define relative blood conductivity $G(t) = \hat{G}(t)/\hat{G}(0)$, $G(t) \in (0, 1]$ and name it blood conductivity degree. Numerically, blood conductivity degree is calculated using formula

$$G(t) = \left( \frac{1}{N} \sum_{i=1}^{N} \frac{1}{k_b(S_i)} \right)^{-1}, \tag{6.3}$$

where $N$ is the number of intervals of a uniform partition of the segment $[0, L]$, $S_i$ is the calculated blood concentration value on $i$th interval. If on some interval $k_b$ is a zero value, then conductivity $G(t)$ is considered equal to zero. Note that any thin region that could be completely occluded will not be stable, not practically nor theoretically (in the model), and breaks up starting from the leading edge due to Oleinik–Liu condition [45].

The reduction of AVM section available for blood flow (increase in degree of AVM section cut off by embolic agent) reduces blood conductivity degree and, therefore, has a positive effect on embolization results. Further, it is illustrated with two examples calculated for patient P Buckley–Leverett function (figure 9). Clinically based AVM geometry parameters are given in appendix A and Buckley–Leverett function construction is given in appendix B.

The first example in figure 3 shows a bad embolization scenario. The boundary conditions (4.3) are: $g^*(t) = (t - 5)/5$, $t_1 = 5$, $t_2 = 10$. It means that only embolic agent enters the AVM input at $t < t_1$. Despite this fact on the embolization front it occupies only part of the cross-section. This is due to the decay of the discontinuity formed at the initial moment, which is unstable according to Oleinik–Liu criterion. Further, at $t \geq t_1$ embolic agent concentration near AVM input starts to decrease and the second discontinuity is formed at $t = t_1$. Embolic agent concentration on embolization front $1 - S \approx 0.15$ remains constant until $t = 13$. Since at this moment posterior discontinuity reaches the anterior one (figure 3b), AVM section cut off by embolic agent begins to decrease monotonically (figure 3c) leading to a significant increase in AVM blood conductivity. In this case, after the end of operation the blood flow through the AVM is largely maintained.

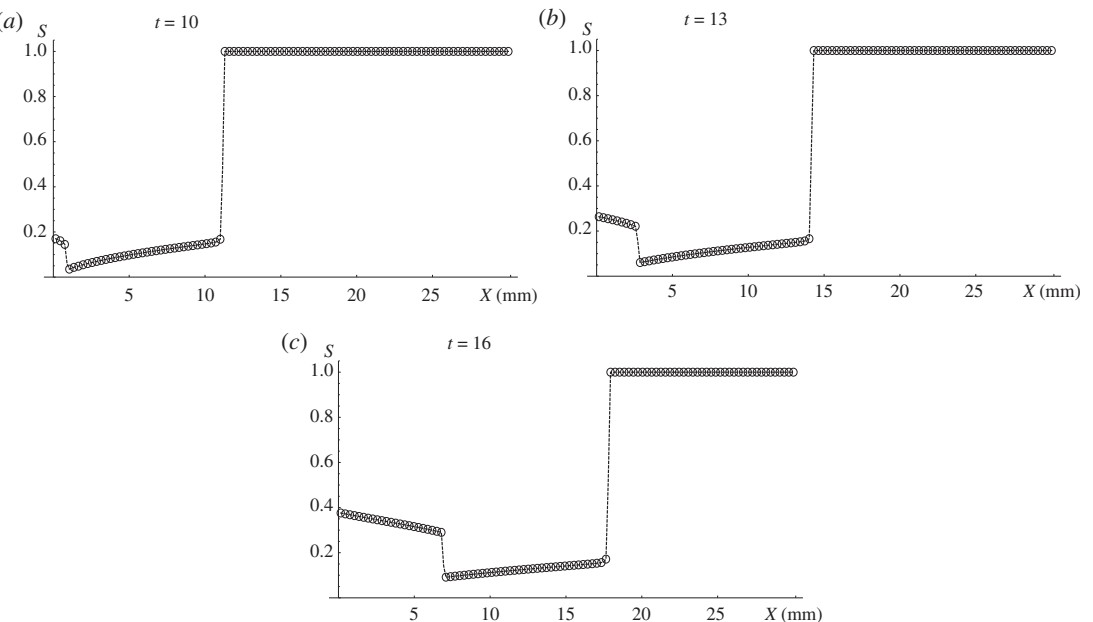

**Figure 4.** Good embolization scenario for patient P. Consecutive stages of the process development at (*a*) *t* = 10, (*b*) *t* = 13, (*c*) *t* = 16. Black dots stand for numerical result and the dashed line is its linear interpolation.

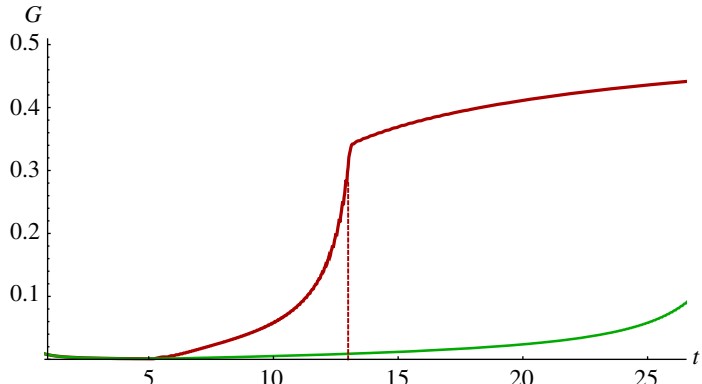

**Figure 5.** Comparison of blood conductivity *G* for patient P: good (lower curve) and bad (upper curve) embolization scenarios. The dashed line indicates the time when two discontinuities merge in a bad scenario.

Figure 4 shows successful (good) embolization scenario. The boundary conditions (4.3) are: $g^*(t) = (t - 5)/30$, $t_1 = 5$, $t_2 = 35$. Here the operation start is the same as in the previous example, but for $t \geq t_1$ the input embolic agent concentration decreases slower than previously. In this case, the posterior discontinuity moves not fast enough to catch up with the anterior one and the embolic agent concentration on embolization front does not decrease. AVM cross-section cut-off will always be not less than on embolization front. In this case, blood conductivity grows much slower over time. Blood conductivity dynamics for both scenarios are presented in figure 5.

These cases demonstrate that operation success depends on embolic agent input $g(t)$, which determines whether AVM cross-section will be cut off by embolic agent the best possible way, or over time a part of AVM section previously blocked by the embolic agent will be reopened for blood flow and blood conductivity will grow.

# 7. Optimal embolization problem

Given the above, the following formulation of the optimal embolization problem as an optimal control problem is proposed:

To minimize the linear combination of AVM blood conductivity degree and AVM fullness by blood at terminal time using embolic agent input as control.

The following conditions must be fulfilled:

(1) Embolization degree should not exceed 60%.
(2) For medical reasons, embolic agent should not reach the AVM exit, that is, it should not enter the vein until the complete blockage of the AVM nidus.
(3) The operation is considered completed at time $T$ when embolic agent ceases to enter the AVM input. In this regard, when modelling only the time interval $[0, T]$ is considered.

In other words, it is necessary to choose a boundary control $g(t)$ such that the solution of problems (2.2), (4.1) and (4.2) delivers minimum to the functional

$$J = \alpha(1 - D(T)) + (1 - \alpha)\, G(T), \quad \alpha \in (0, 1),$$ (7.1)

and satisfies the following conditions:

$$D(t) \leq 0.6, \quad t \in [0, T];$$ (7.2)
$$S(t, L) = 1, \quad t \in [0, T]$$ (7.3)

and

$$S(t, 0) < 1, \quad t \in (0, T).$$ (7.4)

Problem is solved for fixed parameter $\alpha$. Further, this problem for special embolization mode will be considered for specific patients and the results will show that optimal control problem solution delivers minimum for functional for all $\alpha \in (0, 1)$ simultaneously.

## 7.1. Optimal scenario for special embolization case

Let us consider problems (2.2), (4.1) and (4.2) solution for clinically based AVM geometry and Buckley–Leverett functions $f(S)$ (these parameters are given in appendices A and B) and boundary condition of special type given by piecewise-linear function

$$g(t) = \begin{cases} 0, & 0 < t < t_1, \\ (t - t_1)/t_*, & t_1 \leq t < t_2, \\ 1, & t_2 \leq t \leq T, \end{cases}$$ (7.5)

where $t_* = t_2 - t_1$, $t_1 > 0$, $t_2 > t_1$.

This function approximates the natural embolization process: full AVM cross-section cut-off at the beginning with decreasing embolic agent to the end of operation. Thus, in these scenarios embolic agent input is controlled by two parameters, $t_1$ and $t_*$, with end embolization time $T = t_1 + t_*$. Admissible control class (7.5) is extended by $t_* = 0$, meaning that function $g(t)$ has the discontinuity of first type at $t = t_1$

$$g(t) = \begin{cases} 0, & 0 < t < t_1 \\ 1, & t_1 \leq t \leq T \end{cases}, \quad t_* = 0$$ (7.6)

To solve optimal embolization problem for the extended admissible control class (7.5) and (7.6), problems (2.2), (4.1) and (4.2) calculations were made for different values of parameters $t_1$ and $t_*$. Values of $t_1$ vary from 1 to 30 with step 1 while $t_*$ varies from 0.1 to 40.1 with step 1.

The calculations were carried out on a rectangular grid. The spatial coordinate $x$ changes from 0 to $L$ with constant step $h$. Number of space partition points is 200. Note that choice of $L$ does not limit problem generality because it can be set any given value by change of variables in (2.2). Number of time partition points was 2500 at maximum, and time step is given by

$$\tau = \frac{rh}{B}, \quad B = \max_{S \in [0,1]} f'(S),$$ (7.7)

where the Courant number is $r = 0.5$, and Buckley–Leverett function derivative maximum $f'(S)$ is determined individually for each patient. The number of time steps was chosen in such a way that for each patient for all embolization scenarios the embolic agent reached the vein. Further calculations have no practical meaning. Moreover, embolic agent does not exit the malformation then condition (7.2) fulfilment at terminal time $T$ means that it is valid for all times $[0, T]$.

It is convenient to switch to $(t_1, T)$ parameters plane to study optimal embolization problem (figure 6). On this plane, only final states of embolization process are considered. In this case, straight lines parallel to coordinate angle bisector are lines of constant values of $t_* = T - t_1$. The bisector line itself stays for

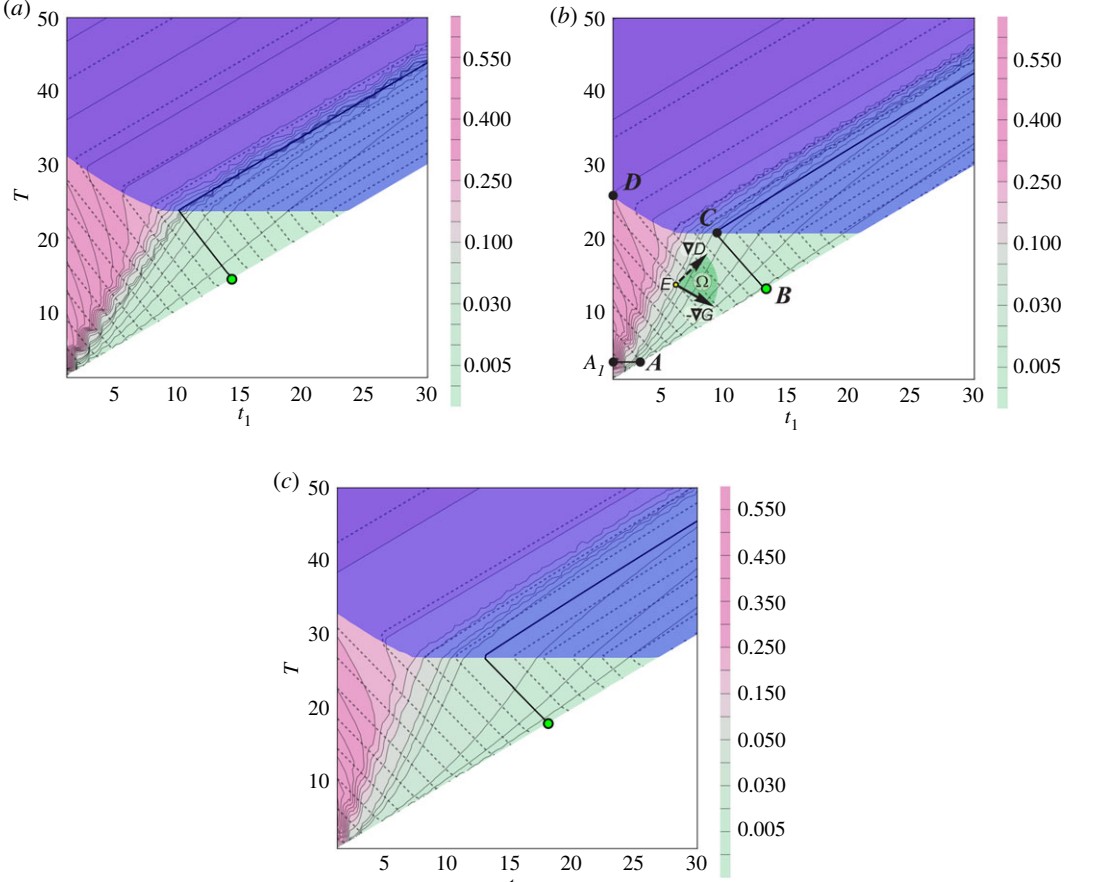

**Figure 6.** $(t_1, T)$ parameters plane for (*a*) patient K, (*b*) patient T, (*c*) patient P. Fill colour determines blood conductivity degree $G(T)$ with level lines shown in solid lines. Dashed lines are level lines of embolization degree $D(T)$ with 60% level shown in bold solid line.

parameter $t_* = 0$ and the area under this line is not considered because of $T < t_1$, which means embolization is finished before time $t_1$ is reached.

Let us analyse optimal embolization problem (7.1)–(7.4) solution construction on the example of patients whose data are shown in figure 9. For this case, plane $(t_1, T)$ is presented in figure 6. In these figures, fill colour determines blood conductivity degree $G(T)$ with level lines shown in solid lines (wiggling is caused by mesh data interpolation). Dashed lines are level lines of embolization degree $D(T)$ with 60% level shown in bold solid line. Points on the right of this line stand for embolization degree greater than 60%. Purple area shows regimes with embolic agent reaching the vein, where condition (7.3) is violated. Since operation duration $T$ is positive (see (4.2)), then a small neighbourhood of the origin should be excluded. Thus for this case, admissible parameters area is $A_1ABCD$. Let us show that for considered patients point $B$, which is the intersection of coordinate angle bisector and 60% embolization degree level line, corresponds to the optimal embolization regime.

Let us minimize objective functional (7.1) using gradient descent method for $\alpha \in (0, 1)$. Objective functional antigradient $-\nabla J = \alpha \nabla D - (1 - \alpha) \nabla G$ is a convex linear combination of blood conductivity degree $G$ and function $1 - D$ antigradients. Figure 6*b* shows that for all $\alpha \in (0, 1)$ antigradient $-\nabla J$ lies between $\nabla D$ and $-\nabla G$ in sector $\Omega$. Location of blood conductivity degree $G$ and embolization degree $D$ level lines in figure 6 illustrates that movement from any interior point $E$ of admissible parameters domain along the objective functional antigradient leads us to the boundary of embolization admissible parameters area $A_1ABCD$ since in a bounded area of admissible parameters the objective functional has no internal extrema. It should be noted that starting from the part of the boundary $A_1D$, we immediately go inside the region of admissible parameters, since objective functional decreases with fixed $T$ while parameter $t_1$ grows. Movement along the part of boundary $\Gamma = A_1ABCD \setminus A_1D$ towards functional descending leads us to point $B$, which delivers objective functional minimum. Note that $\nabla G$ on boundary part $AB$ and $\nabla D$ on boundary part $BC$ projections are empty, since $AB$ and $BC$ are $G$ and $D$ level lines, respectively. But since $\alpha \in (0, 1)$ then objective functional gradient projection on the boundary is non-empty on $\Gamma$.

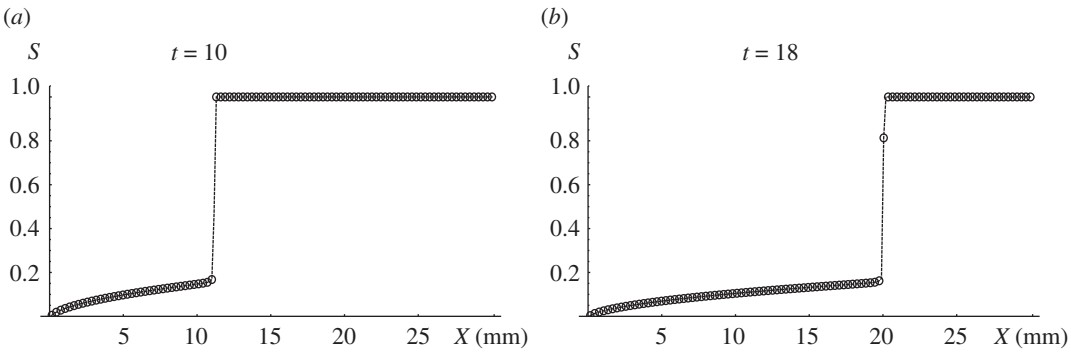

**Figure 7.** Optimal solution for patient P which has initial discontinuity decays in consecutive times; (a) $t = 10$, (b) $t = 18$.

The found optimal embolization regime, given by point **B**, is discontinuous and corresponds to the function (7.6). At $t = 0$, there is no embolic agent in AVM inlet, at $t \in (0, T)$ AVM inlet is completely blocked by embolic agent, and at terminal time $t = T$ embolization is completed and there is no more embolic agent in AVM inlet. At the end of operation time $T$ embolization degree is 60%. Figure 7 shows corresponding solution for patient P in two consecutive times.

Figure 6 shows that for all patients of concern following simple qualitative conditions are valid:

(1) inside the area $A_1ABCD$ objective functional decreases with fixed $T$ while parameter $t_1$ grows and has no internal minima;
(2) functional decreases on the boundary while moving along its parts $A_1AB$ and $DCB$ towards point **B**;
(3) embolic agent does not reach the vein during operation with parameter values corresponding to point **B**.

It is easy to prove that these conditions are sufficient for optimal embolization regime to correspond to the same point **B**.

The course of embolization is determined by AVM length $L$, Buckley–Leverett function $f(S)$, which describes malformation properties, and control $g(t)$, which sets embolization regime. Qualitative character of conditions (1)–(3) entail that for Buckley–Leverett functions close to considered embolization regimes will be similar. In particular, it means that Buckley–Leverett function construction from clinical data could be made with sufficiently big error. This conditions will also be valid for close to considered piecewise-linear and nonlinear classes of control function $g(t)$. For all these classes, optimal embolization regime will also lie on the intersection of the main diagonal and level line of maximal admissible embolization degree.

Generally speaking, the situation when embolic agent reaches the vein earlier than 60% embolization degree is achieved is not excluded. In this case, boundary $A_1ABCD$ part $BC$ disappears and optimal regime lies on the intersection of purple area boundary and main diagonal and also is discontinuous. This case did not appear among the considered patients.

Practically, it is impossible to maintain optimal regime exactly. But since objective functional value varies insignificantly in the neighbourhood of point **B** then regime from the neighbourhood of point **B** could be quite acceptable.

## 8. Discussion

In this paper, optimal AVM embolization scenario in terms of safety and effectiveness was studied. Joint blood and embolic agent flow in AVM nidus was described using one-dimensional two-phase filtration model. For simulation, Godunov's monotonic modification of CABARET scheme was used, which allows calculation of discontinuous solutions for scalar conservation law with a non-convex flow. Absolute permeabilities for intact malformations were determined for real patients *in vivo*. Also, Buckley–Leverett functions for malformations were constructed based on the clinical data of real patients, and were used to study the optimal embolization problem. AVM optimal embolization problem was formulated, its numerical solution for special linear embolization regime was constructed, and admissible and optimal embolization scenarios were calculated. The optimal scenario for all examined clinical cases corresponds to a discontinuous embolization regime. Namely, complete closure of AVM

inlet cross-section with an embolic agent, then bringing the embolic agent amount in the AVM nidus to admissible maximum and a sharp cessation of embolic agent delivery.

Available methods for AVM geometry reconstruction from neuroimaging data (CT, MRI, cerebral angiography) make it possible to distinguish vessels with an average diameter of at least 0.5 mm. It is insufficient to determine *in vivo* the detailed geometric AVM structure, which consists of a very large number of intertwined thin vessels whose diameter can reach up to 0.1 mm. Therefore, when modelling AVMs, it becomes necessary to use simplified approaches. The paper considers a one-dimensional model of AVM embolization, which allows us to describe important qualitative patterns of this process. In addition to the difficulty of AVM geometry reconstruction, the mathematical description of blood rheology, arterial and venous vessels wall properties, the forces arising in vessels walls, as well as the forces acting on the vessel from the surrounding tissues, the filtration processes through the vessel wall, chemical reactions and other ongoing processes are still far away from completion. Despite this, taking into account the most significant and determining parameters of the system allows one to build mathematical models that give a qualitatively correct description of its behaviour.

For a more detailed study of embolization in future, it is necessary to consider more accurate models that take into account additional characteristics of the AVM embolization process. This primarily concerns the uneven distribution of the physical characteristics of the AVM (porosity, permeability and cross-sectional area) along model length. More accurate embolic agent behaviour description is also important, since some of its varieties are liquids with pronounced non-Newtonian properties. In addition, embolic agent adsorption in blood vessels walls and its properties variation over time during solidification are of interest. Despite the impossibility at present of complete AVM internal structure restoration using neuroimaging data, macroscopic characteristics such as the shape of cross-sections and the length of AVM nidus can be restored with sufficient for modelling accuracy. In this regard, the next stage of the study may be the consideration of two- and three-dimensional AVM models. To solve the stated modelling problems, further testing, development and improvement of the used numerical CABARET method is necessary.

Data accessibility. Electronic supplementary materials include tables of clinical data of patients obtained during neurosurgical operations to cure vascular pathologies. It should be noted that pressure measurements in the vein are difficult to implement, as a result of which literary data on the pressure on the venous part of the pathology were used in the study. At the same time, for six patients, pressure was measured in the vein before surgery, which allowed us to estimate the order of absolute permeability magnitude. Materials also include the description and pseudo-code of the numerical CABARET method used to obtain the results of the study.

Authors' contributions. A.A.C. carried out the development of the work concept, obtaining clinical data, the construction of a mathematical model, writing program code, obtaining the numerical results and their interpretation. T.S.G. participated in the construction of a mathematical model, in writing program code, obtaining numerical results and their interpretation. I.A.P. participated in the construction of a mathematical model and accompanied the work from the point of view of optimal control theory. V.V.O. carried out the development of a numerical scheme and interpretation of numerical results. V.A.P. accompanied the work from the point of view of medicine and carried the clinical data obtaining.

Competing interests. We declare we have no competing interest.

Funding. This work was supported by the Russian Science Foundation (grant no. 16-11-10033) (numerical methods and calculations); and by the Government of the Russian Federation (grant no. 14.W03.31.0002) (medical data and embolization problem).

# Appendix A. Clinically based arteriovenous malformation geometry and permeabilities

Philips ComboMap system and the Philips ComboWire sensor (sensor diameter is 0.36 mm and its length is 1.85 m) were used in neurosurgical operations to measure blood velocity and pressure inside the cerebral vessels near the pathology. This way the values of pressure and velocity $v_b$ in AVM inlet (artery) were obtained before, during and after the embolization for 10 patients. Further data on geometrical AVM parameters such as length $L$ and cross-sectional area $A$ as well as inlet artery cross-sectional area $\omega$ are used. These parameters are obtained based on perioperative X-ray tomography. Further three patients $K$, $T$ and $P$ with the largest number of intraoperative measurements are considered.

From (2.4), one can see that Buckley–Leverett function is completely determined by the relative phase permeabilities and viscosities of the blood and embolic agent. Known blood viscosity [11] and common

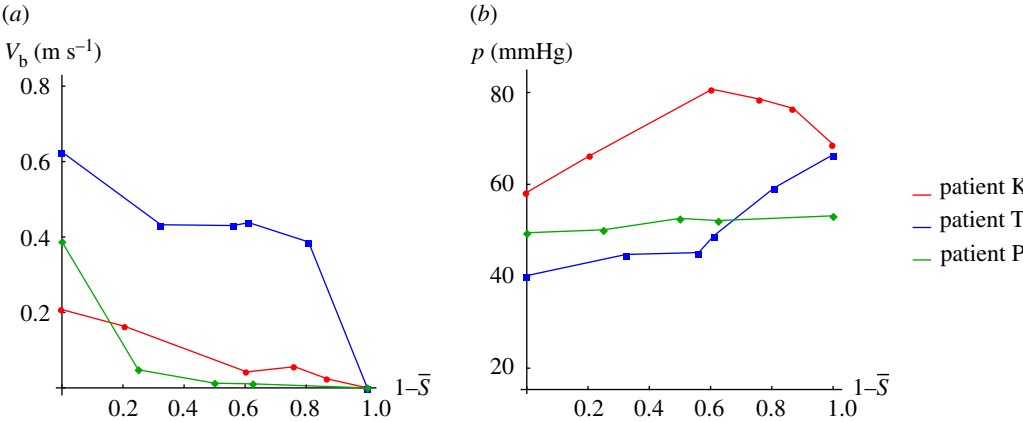

**Figure 8.** Clinical data on blood velocity and pressure in arterial AVM inlet versus embolic agent fraction in the AVM.

**Table 1.** AVM parameters.

| patient | $L$ ($10^{-2} \times$ m) | $A$ ($10^{-4} \times$ m$^2$) | $\omega$ ($10^{-6} \times$ m$^2$) | $K$ (m$^2$) |
|---------|------|------|------|------|
| K | 2.4 | 4.5 | 4.5 | $8.2 \times 10^{-11}$ |
| T | 2.2 | 2.0 | 4.5 | $6.7 \times 10^{-8}$ |
| P | 3.0 | 3.1 | 3.1 | $3.7 \times 10^{-10}$ |

embolic agent ONYX18 [7] are

$$\eta_b \approx 4\,cP \quad \text{and} \quad \eta_e \approx 18\,cP. \tag{A 1}$$

To calculate blood relative phase permeability, its absolute and phase permeabilities are needed. Absolute permeability $K$ is porous medium (AVM nidus) characteristic. It is assumed to be constant and could be calculated using monitoring data before embolization while embolic agent is not in the AVM and it is completely filled with blood. From Darcy's Law, we get

$$K = -\frac{Q_{b_0}\,\eta_b}{A(\Delta p_0/L)} \equiv \text{const.}, \tag{A 2}$$

where $Q_{b_0} = v_{b_0}\,\omega$ is blood volume flow before embolization, $v_{b_0}$, blood flow velocity in the arterial AVM inlet before embolization, $\Delta p_0$, pressure drop between arterial AVM inlet and venous AVM outlet before embolization. Values $Q_{b_0}$ and $\Delta p_0$ are based on monitoring data.

Total AVM embolization was achieved in three considered operations, assuming that in total embolization the AVM nidus is filled by embolic agent completely. Monitoring data include velocity, pressure and injected embolic agent fraction. Using information on the total amount of embolic agent used for the operation and its fractions in the intermediate time points $t_i$ we can calculate volume average blood concentration $\bar{S}(t_i)$ in the AVM.

Assuming that embolic agent is always distributed uniformly among AVM blood phase permeability $K_b$ at time $t_i$ is given by formula

$$K_b(\bar{S}_i) = -\frac{Q_b(\bar{S}_i)\,\eta_b}{A\,(\Delta p(\bar{S}_i)/L)}, \tag{A 3}$$

where $\bar{S}_i = \bar{S}(t_i)$, $Q_b(\bar{S}_i)$ is blood flow and $\Delta p(\bar{S}_i)$ is pressure drop between artery and vein during embolization. Values $Q_b(\bar{S}_i) = v_b(\bar{S}_i)\,\omega$ and $\Delta p(\bar{S}_i)$ are also calculated using monitoring data.

Then blood relative phase permeability $k_b(\bar{S}_i)$ is the following:

$$k_b(\bar{S}_i) = \frac{K_b(\bar{S}_i)}{K} = \frac{v_b(\bar{S}_i)\,\Delta p_0}{v_{b_0}\,\Delta p(\bar{S}_i)}. \tag{A 4}$$

Formula (A 4) shows that $k_b(\bar{S}_i)$ is independent of blood and embolic agent viscosity and AVM geometric properties.

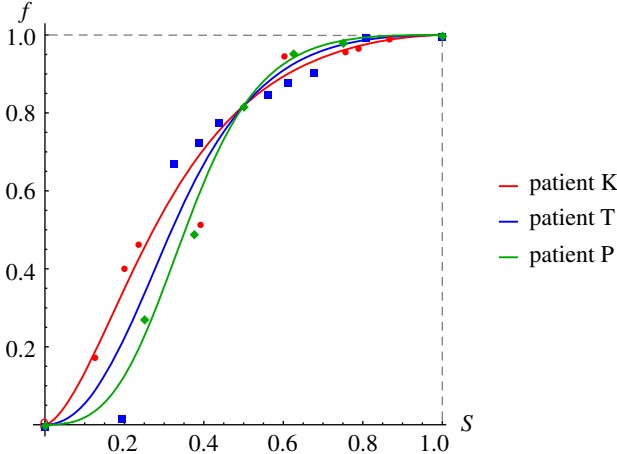

**Figure 9.** Buckley–Leverett functions constructed using clinical data: circles for patient K, squares for patient T, diamonds for patient P. Lines present Corey model approximations.

**Table 2.** Corey model parameter.

| patient | $\kappa$ | data s.e. |
| --- | --- | --- |
| K | 1.54 | ±0.026 |
| T | 1.99 | ±0.036 |
| P | 2.49 | ±0.018 |

Figure 8 shows monitoring data on arterial velocity and pressure [47] during embolization for patients $K$, $T$ and $P$. Pressure increases and velocity decreases during embolization. Venous pressure is assumed to decrease linearly during embolization from 40 to 7 mmHg [48].

AVM geometrical parameters for three considered patients were determined based on intraoperative X-ray angiography and perioperative X-ray tomography and are presented in table 1.

Absolute permeability values used for further calculations are presented in table 1. For patient T with permeability two orders greater, the pressure drop at the AVM inlet and outlet is less than the drop for patients K and P. Currently, available clinical measurements for 10 patients with cerebral AVM allow us to assume that absolute permeability is within $10^{-8}$–$10^{-12}$ m² and for seven patients is of order $10^{-10}$. As far as authors know, previously value of absolute permeability *in vivo* for such inaccessible biological objects as AVM was not presented. This bio-mechanical AVM nidus porous medium characteristic was obtained due to *in vivo* velocity and pressure measurements.

# Appendix B. Clinically based Buckley–Leverett function construction

For further calculations, the Corey model [49] was used for the analytic approximation of blood relative phase permeability

$$k_b(S) = S^\alpha, \quad \alpha \in \mathbb{R}, \quad \alpha > 1. \tag{B 1}$$

The embolic agent relative permeability $k_e(S)$ is assumed to be symmetric to blood relative permeability in the following sense:

$$k_e(S) = k_b(1 - S). \tag{B 2}$$

Thus, Buckley–Leverett function is the following:

$$b(S) = \frac{S^\kappa / \eta_b}{S^\kappa / \eta_b + (1 - S)^\kappa / \eta_e}, \tag{B 3}$$

where coefficient $\kappa$ was calculated using the least-square method for clinical data. This type of Buckley–Leverett function guarantees its non-convexity and fulfilment of condition (2.5).

Buckley–Leverett function approximations were constructed using clinical data for patients of concern. Table 2 shows the obtained Corey model parameter values and standard error (s.e.) of data recovery.

In figure 9, markers stand for Buckley–Leverett function values obtained by formulae (2.4), (A 4) and (B 2) using clinical data, and lines show Buckley–Leverett approximations by formula (B 3).

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
