## [Reviewer comments · Royal Society Open Science]

Review History

RSOS-191992.R0 (Original submission)

Review form: Reviewer 1

Is the manuscript scientifically sound in its present form?

Yes

Are the interpretations and conclusions justified by the results?

Yes

Is the language acceptable?

No

Do you have any ethical concerns with this paper?

No

Have you any concerns about statistical analyses in this paper?

No

Recommendation?

Major revision is needed (please make suggestions in comments)

Comments to the Author(s)

I think this is a really interesting paper with a novel simplified approach to understanding and optimising the embolisation process. The discussion regarding the overtake of one discontinuity by another and optimal approach seem very insightful. However, I think some revisions are required and some concerns need to be addressed before I would be happy to recommend publication.

comments/suggestions:

the paper is poorly written in some places and as a result the reasoning remains somewhat unclear (particularly section 4 and section 7).

The authors quite rightly acknowledge at the end of section 8 that more detailed studies are required that include aspects such as embolic agent behaviour, spatial characteristics of AVM, blood rheology and wall properties. My feeling though is that some of this should be mentioned in the introduction - particularly concerning what is assumed regarding how blood and different embolic agents interact and if that is realistic for embolic agents in use.

I am marginally unconvinced regarding removing the cardiac cycle effect given the typical timescale of injection and the possible large fluctuations in pressure and flow rate so perhaps some further comment on that is necessary.

The explanation regarding the incoming blood concentration in equations (6) and (7) is unclear - please rewrite and explain better.

Equation (13) - should this have a $1/L$ in front of the integral?

Equation (14) - I am a little concerned about the following comment below this equation: "if on some interval k_b is a zero value then the conductivity $G(t)$ is considered equal to zero" - could this be troubling from a grid dependence perspective? My point is that perhaps during/after the process a very thin region may become completely occluded in the model but this region is highly unlikely to remain stable post surgery and will be considered a failed embolisation.

Section 7 - I found section 7.1 incredibly unclear and difficult to follow - please rewrite. In the first paragraph of p14 please just explain better (or properly highlight in the figure) the line where $t^*=0$. In the second paragraph what is the purple area? And what do you mean by "obvious medical condition" (line 21 p 14)? I understand T cannot be zero but I am missing something?

Figure 6 -The solid contour lines that go from the origin to near the corner point of the 60% contour line display very significant wiggling - is this a problem with the numerics? An explanation should be mentioned in the text.

Decision letter (RSOS-191992.R0)

18-Mar-2020

Dear Dr Cherevko,

The editors assigned to your paper ("Modeling of the arteriovenous malformation embolization optimal scenario") have now received comments from reviewers. We would like you to revise

your paper in accordance with the referee and Associate Editor suggestions which can be found below (not including confidential reports to the Editor). Please note this decision does not guarantee eventual acceptance.

Please submit a copy of your revised paper before 10-Apr-2020. Please note that the revision deadline will expire at 00.00am on this date. If we do not hear from you within this time then it will be assumed that the paper has been withdrawn. In exceptional circumstances, extensions may be possible if agreed with the Editorial Office in advance. We do not allow multiple rounds of revision so we urge you to make every effort to fully address all of the comments at this stage. If deemed necessary by the Editors, your manuscript will be sent back to one or more of the original reviewers for assessment. If the original reviewers are not available, we may invite new reviewers.

- Data accessibility

<http://datadryad.org/submit?journalID=RSOS&manu=RSOS-191992>

- Competing interests

- Authors' contributions

All submissions, other than those with a single author, must include an Authors' Contributions section which individually lists the specific contribution of each author. The list of Authors

should meet all of the following criteria; 1) substantial contributions to conception and design, or acquisition of data, or analysis and interpretation of data; 2) drafting the article or revising it critically for important intellectual content; and 3) final approval of the version to be published.

- Acknowledgements

- Funding statement

on behalf of Dr Oliver Jensen (Associate Editor) and Mark Chaplain (Subject Editor)
openscience@royalsociety.org

Associate Editor's comments (Dr Oliver Jensen):

Comments to the Author:

Please revise your paper in line with the comments from the reviewer, being careful to address every point in full. There are elements of the presentation that are unclear, where further explanation would be helpful, as the reviewer has indicated. In addition, there are some aspects of the writing that could be clarified. Specifically, "nidus" will be an unfamiliar term to most readers that needs explanation when it is introduced, "eloquent" (page 2) seems inappropriate, as does "high request" (page 2) and "income" (page 14). Please ensure that the physical assumptions underlying the transport equation (1) are explained fully, perhaps by presenting the underlying model using physical time, and define clearly the "energy" of the flow (page 7). Finally, retitle Section 8 as "Discussion," as results have been presented previously.

Reviewers' Comments to Author:

Reviewer: 1

Comments to the Author(s)

I think this is a really interesting paper with a novel simplified approach to understanding and optimising the embolisation process. The discussion regarding the overtake of one discontinuity by another and optimal approach seem very insightful. However, I think some revisions are

required and some concerns need to be addressed before I would be happy to recommend publication.

comments/suggestions:

the paper is poorly written in some places and as a result the reasoning remains somewhat unclear (particularly section 4 and section 7).

The authors quite rightly acknowledge at the end of section 8 that more detailed studies are required that include aspects such as embolic agent behaviour, spatial characteristics of AVM, blood rheology and wall properties. My feeling though is that some of this should be mentioned in the introduction - particularly concerning what is assumed regarding how blood and different embolic agents interact and if that is realistic for embolic agents in use.

I am marginally unconvinced regarding removing the cardiac cycle effect given the typical timescale of injection and the possible large fluctuations in pressure and flow rate so perhaps some further comment on that is necessary.

The explanation regarding the incoming blood concentration in equations (6) and (7) is unclear - please rewrite and explain better.

Equation (13) - should this have a $1/L$ in front of the integral?

Equation (14) - I am a little concerned about the following comment below this equation: "if on some interval k_b is a zero value then the conductivity $G(t)$ is considered equal to zero" - could this be troubling from a grid dependence perspective? My point is that perhaps during/after the process a very thin region may become completely occluded in the model but this region is highly unlikely to remain stable post surgery and will be considered a failed embolisation.

Section 7 - I found section 7.1 incredibly unclear and difficult to follow - please rewrite. In the first paragraph of p14 please just explain better (or properly highlight in the figure) the line where $t^*=0$. In the second paragraph what is the purple area? And what do you mean by "obvious medical condition" (line 21 p 14)? I understand T cannot be zero but I am missing something?

Figure 6 -The solid contour lines that go from the origin to near the corner point of the 60% contour line display very significant wiggling - is this a problem with the numerics? An explanation should be mentioned in the text.

Author's Response to Decision Letter for (RSOS-191992.R0)

See Appendix A.

RSOS-191992.R1 (Revision)

Review form: Reviewer 1

Is the manuscript scientifically sound in its present form?

Yes

Are the interpretations and conclusions justified by the results?

Yes

Is the language acceptable?

No

Do you have any ethical concerns with this paper?

No

Have you any concerns about statistical analyses in this paper?

No

Recommendation?

Accept with minor revision (please list in comments)

Comments to the Author(s)

The authors have addressed the majority of my comments and so I am happy to recommend publication provided the language is improved. I cannot list all potential typos/errors but here are some examples/suggestions:

using "than" instead of "then"

Just above equation (1) "Denote local.... " -> "We denote the local... "

Just above equation (2) " Under this assumptions..."

Missing articles throughout

for example - just above equation (7) ", determined by operation strategy" "determined by the chosen clinical/embolisation strategy"

Decision letter (RSOS-191992.R1)

Dear Dr Cherevko,

On behalf of the Editors, I am pleased to inform you that your Manuscript RSOS-191992.R1 entitled "Modeling of the arteriovenous malformation embolization optimal scenario" has been accepted for publication in Royal Society Open Science subject to minor revision in accordance with the referee suggestions. Please find the referees' comments at the end of this email.

The reviewers and Subject Editor have recommended publication, but also suggest some minor revisions to your manuscript. Therefore, I invite you to respond to the comments and revise your manuscript.

- Ethics statement

- Data accessibility

<http://datadryad.org/submit?journalID=RSOS&manu=RSOS-191992.R1>

- Competing interests

- Authors' contributions

- Acknowledgements

- Funding statement

Because the schedule for publication is very tight, it is a condition of publication that you submit the revised version of your manuscript before 28-Jun-2020. Please note that the revision deadline will expire at 00.00am on this date. If you do not think you will be able to meet this date please let me know immediately.

To revise your manuscript, log into <https://mc.manuscriptcentral.com/rsos> and enter your Author Centre, where you will find your manuscript title listed under "Manuscripts with Decisions". Under "Actions," click on "Create a Revision." You will be unable to make your

revisions on the originally submitted version of the manuscript. Instead, revise your manuscript and upload a new version through your Author Centre.

Kind regards,
Lianne Parkhouse
Editorial Coordinator
Royal Society Open Science
openscience@royalsociety.org

on behalf of Dr Oliver Jensen (Associate Editor) and Mark Chaplain (Subject Editor)
openscience@royalsociety.org

Associate Editor Comments to Author (Dr Oliver Jensen):

Please ensure that remaining grammatical errors are fixed. The referee has identified a handful of them but there are others that need to be addressed also.

Reviewer comments to Author:

Reviewer: 1

Comments to the Author(s)

The authors have addressed the majority of my comments and so I am happy to recommend publication provided the language is improved. I cannot list all potential typos/errors but here are some examples/suggestions:

using "than" instead of "then"

Just above equation (1) "Denote local.... " -> "We denote the local.. "

Just above equation (2) " Under this assumptions..."

Missing articles throughout

for example - just above equation (7) ", determined by operation strategy" "determined by the chosen clinicail/embolisation strategy"

Author's Response to Decision Letter for (RSOS-191992.R1)

See Appendix B.

Decision letter (RSOS-191992.R2)

Dear Dr Cherevko,

It is a pleasure to accept your manuscript entitled "Modeling of the arteriovenous malformation embolization optimal scenario" in its current form for publication in Royal Society Open Science.

We note that the following email address appears to be incorrect: ostapenko@hydro.nsc.ru

Please check this for your co-authors, and kindly respond to this email with the correct email address for this author.

Due to rapid publication and an extremely tight schedule, if comments are not received, your paper may experience a delay in publication. Royal Society Open Science operates under a continuous publication model. Your article will be published straight into the next open issue and this will be the final version of the paper. As such, it can be cited immediately by other

researchers. As the issue version of your paper will be the only version to be published I would advise you to check your proofs thoroughly as changes cannot be made once the paper is published.

Best regards,

on behalf of Dr Oliver Jensen (Associate Editor) and Mark Chaplain (Subject Editor)
openscience@royalsociety.org

Appendix A

Associate Editor's comments (Dr Oliver Jensen):

Comments to the Author:

Please revise your paper in line with the comments from the reviewer, being careful to address every point in full. There are elements of the presentation that are unclear, where further explanation would be helpful, as the reviewer has indicated. In addition, there are some aspects of the writing that could be clarified.

Dr Oliver Jensen: Specifically, "nidus" will be an unfamiliar term to most readers that needs explanation when it is introduced,

Authors: Authors explained the word "nidus" in introduction.

Dr Oliver Jensen: "eloquent" (page 2) seems inappropriate, as does "high request" (page 2) and "income" (page 14).

Authors: The part of the article with these words was reviewed.

Dr Oliver Jensen: Please ensure that the physical assumptions underlying the transport equation (1) are explained fully, perhaps by presenting the underlying model using physical time, and define clearly the "energy" of the flow (page 7).

Authors: In the revision equation in physical terms is given before the transition to simplified form of equation in modified time is explained in detail. (page 4)

Formulas and explanations defining energy of the flow are added to section 5 (now page 8).

Dr Oliver Jensen: Finally, retitle Section 8 as "Discussion," as results have been presented previously.

Authors: Done.

Reviewers'(R) Comments to Author(A):

R: the paper is poorly written in some places and as a result the reasoning remains somewhat unclear (particularly section 4 and section 7).

Authors: For better understanding of the article, the authors have slightly changed chapters 4 and 7.

R: The authors quite rightly acknowledge at the end of section 8 that more detailed studies are required that include aspects such as embolic agent behavior, spatial characteristics of AVM, blood rheology and wall properties. My feeling though is that some of this should be mentioned in the introduction - particularly concerning what is assumed regarding how blood and different embolic agents interact and if that is realistic for embolic agents in use.

Authors: All assumptions made by authors are presented in the Introduction of the revised article. Constructed model is quite simplified and considers main feature of blood and embolic agent interaction as Newtonian liquids with constant viscosities. On this stage the model neglects other interaction mechanisms.

R: I am marginally unconvinced regarding removing the cardiac cycle effect given the typical timescale of injection and the possible large fluctuations in pressure and flow rate so perhaps some further comment on that is necessary.

Authors: For various cardiac cycle types with different forms of volumetric flow rate transition to modified time allows to get a same equation with unit volumetric flow rate. Inverse time modification leads to the solution of equation corresponding to physical time-flow relation. Moreover, according to authors' observations during neurosurgical operations, pressure and blood flow fluctuations during the cardiac cycle do not cause significant changes of geometrical AVM parameters.

R: The explanation regarding the incoming blood concentration in equations (6) and (7) is unclear - please rewrite and explain better.

Authors: Now equations (6), (7) are (7), (8). The indicated fragment was rewritten for clarification.

R: Equation (13) - should this have a $1/L$ in front of the integral?

Authors: In equation (14) (previously (13)) function $R(t, x)$ is a local unit resistance. Thus multiplier $1/L$ is excess that is clarified in the text.

R: Equation (14) - I am a little concerned about the following comment below this equation: "if on some interval k_b is a zero value then the conductivity $G(t)$ is considered equal to zero" - could this be troubling from a grid dependence perspective? My point is that perhaps during/after the process a very thin region may be become completely occluded in the model but this region is highly unlikely to remain stable post surgery and will be considered a failed embolisation.

Authors: Typical AVM size is several centimeters. During numerical calculations 200 evenly distributed in length points are used. Thus, spatial resolution is fractions of millimeter which is enough for grid convergence and, in our opinion, is quiet enough for embolization process description. If such a thin region occurs, it will not be stable in the model and breaks up starting from the leading edge due to Oleinik-Liu condition, similarly to situation shown on figures 3 and 4 (centered rarefaction wave adjacent to right gap to the left).

R: Section 7 - I found section 7.1 incredibly unclear and difficult to follow - please rewrite. In the first paragraph of p14 please just explain better (or properly highlight in the figure) the line where $t^=0$. In the second paragraph what is the purple area? And what do you mean by "obvious medical condition" (line 21 p 14)? I understand T cannot be zero but I am missing something?*

Authors: This fragment has been completely rewritten in order to clarify the presentation of material. Line where $t^*=0$ corresponds to the discontinuous embolization mode (formula (21)) and in the figure it is the boundary of the region under consideration (bisector of parameter plane $T-t_1$). We changed the name of the area from blue to purple, which is more consistent with the color of this area in the picture. The only constraint implied by "obvious medical condition" is $T>0$.

R: Figure 6 -The solid contour lines that go from the origin to near the corner point of the 60% contour line display very significant wiggling - is this a problem with the numerics? An explanation should be mentioned in the text.

Authors: Blood conductivity level lines are shown in solid lines and their wiggling is caused by mesh data interpolation. It is mentioned in the revision.

Modeling of the arteriovenous malformation embolization optimal scenario

Abstract

Cerebral Arterio-Venous Malformation (AVM) is a congenital brain vessels pathology, in which the arterial and venous blood channels are connected by a nidus of randomly interwoven degenerate vessels. It is a dangerous disease that affects brain functioning causing the high risk of intracerebral hemorrhage. One of AVM treatment methods is embolization – the endovascular filling of the AVM vessel bundle with a special embolic agent. This method is widely used, but still in some cases is accompanied by intraoperative AVM vessels rupture. In this paper the optimal scenario of AVM embolization is studied from the point of view of safety and effectiveness of the procedure. The co-movement of blood and embolic agent in the AVM body is modeled on the basis of a one-dimensional two-phase filtration model. Optimal control problem with phase constraints arising from medicine is formulated and numerically solved. In numerical analysis, the monotone modification of the CABARET scheme is used. Optimal embolization model is constructed on the basis of real patients clinical data collected during neurosurgical operations. For the special case of embolic agent input admissible and optimal embolization scenarios were calculated.

Keywords

Hemodynamics, arteriovenous malformation, optimal control, two-phase filtration, CABARET scheme

1 Introduction

Cerebral arteriovenous malformation (AVM) is a congenital abnormality of brain vessels, in which a direct discharge of blood from the arterial blood pool into the venous pool is present, bypassing the capillary vessel network [1], [2], [3], [4], [5]. Due to capillary network absence, the resistance of the corresponding part of circulatory system decreases. This deviations in vascular system modify both hemodynamic parameters (flow velocity and pressure) and strength properties of blood vessels. Most often, AVMs become symptomatic at the age of 20 - 40 years and are detected during this period of life. Moreover, in more than half cases the disease manifests itself by hemorrhage. Disability related to hemorrhage from AVM reaches 50% with a risk of death of 10-29%. For AVM posterior cranial fossa the mortality rate exceeds 50%.

The hemodynamic associated with intracranial AVMs is complex and vary as morphology and angioarchitecture changes, especially with different treatment methods. Various treatment methods effects on AVM and its environment hemodynamics are presented, for example, in the review of [6].

Modeling of the arteriovenous malformation embolization optimal scenario

Abstract

Cerebral Arterio-Venous Malformation (AVM) is a congenital brain vessels pathology, in which the arterial and venous blood channels are connected by tangles of abnormal blood vessels. It is a dangerous disease that affects brain functioning causing the high risk of intracerebral hemorrhage. One of AVM treatment methods is embolization – the endovascular filling of the AVM vessel bundle with a special embolic agent. This method is widely used, but still in some cases is accompanied by intraoperative AVM vessels rupture. In this paper the optimal scenario of AVM embolization is studied from the point of view of safety and effectiveness of the procedure. The co-movement of blood and embolic agent in the AVM body is modeled on the basis of a one-dimensional two-phase filtration model. Optimal control problem with phase constraints arising from medicine is formulated and numerically solved. In numerical analysis, the monotone modification of the CABARET scheme is used. Optimal embolization model is constructed on the basis of real patients clinical data collected during neurosurgical operations. For the special case of embolic agent input admissible and optimal embolization scenarios were calculated.

Keywords

Hemodynamics, arteriovenous malformation, optimal control, two-phase filtration, CABARET scheme

1 Introduction

Cerebral arteriovenous malformation (AVM) is a congenital abnormality of brain vessels, in which a direct discharge of blood from the arterial blood pool into the venous pool is present, bypassing the capillary vessel network [1], [2], [3], [4], [5]. Due to capillary network absence, the resistance of the corresponding part of circulatory system decreases. This deviations in vascular system modify both hemodynamic parameters (flow velocity and pressure) and strength properties of blood vessels. Most often, AVMs become symptomatic at the age of 20 - 40 years and are detected during this period of life. Moreover, in more than half cases the disease manifests itself by hemorrhage. Disability related to hemorrhage from AVM reaches 50% with a risk of death of 10-29%. For AVM posterior cranial fossa the mortality rate exceeds 50%.

The hemodynamic associated with intracranial AVMs is complex and vary as morphology and angioarchitecture changes, especially with different treatment methods. Various treatment methods effects on AVM and its environment hemodynamics are presented, for example, in the review of [6].

At the moment the most common methods of AVM treatment are microsurgical, endovascular and radiosurgical. Endovascular embolization is the selective deactivation of blood vessels from the bloodstream by filling them with a special embolic agent. Modern level of medicine development makes embolization the most preferable method due to minimal invasiveness and possibility of operating in the deep, eloquent areas of the brain [7]. This method is widely used. Despite the well-developed embolization technique, the risk of intraoperative vascular rupture remains a concern. According to a study of 408 patients with AVM who were treated endovascular, it was noted 11 % frequency of hemorrhagic complications associated with the treatment [8]. Thus, modeling of AVM embolization is of high request.

By type of vessels, connecting the arterial and venous basins, AVMs can be divided into fistula (consisting of large vessels) and racemic (consisting of large number of small diameter vessels, which are chaotically intertwined and intersect each other). There are also mixed-type AVMs that combine both racemic and fistula parts. Modern medical examination methods such as CT, MRI and cerebral angiography) do not allow to determine the detailed geometric AVM structure in vivo since its nidus consists of very large number of complexly interwoven thin vessels with diameter down to 0.1 mm which exceeds medical equipment resolution. Therefore, various simplified methods for AVM modeling are needed.

Mathematical description of blood flow is often based on one-dimensional approximation of Navier-Stokes equations by averaging over blood vessel cross section. Analytical studies of equations describing flows in pipes with rigid and elastic boundaries can be found in the works of [9], [10], [11]. Viscous fluid flow in a network of soft tubes is modeled in the works of [12], [13], [14] on the base of one-dimensional approximation of mass and momentum conservation laws. Numerical simulation of hemodynamics for large blood vessels based on 3D-1D coupled flow is considered, for example, in the works of [15], [16], [17].

AVM and blood flow in surrounding vessels flow interaction is often studied in analogy with electric and hydraulic networks [18], [19], [20]. Such models make it possible to evaluate the effect of various embolization scenarios on blood flow restructuration and are consistent with the general medical view on AVM hemodynamics. The effect of AVM on the blood flow in the surrounding vessels has also been studied using the mass and momentum conservation laws of the ideal incompressible fluid flow. The resulting equations form a hyperbolic system of differential conservation laws. One dimensional variant of this system on a vessel graph was studied in the works of [21], [22] and its two dimensional variant on branching network channels was considered in the works of [23], [24]. The process of embolization was studied in the work of [25] where two-phase flow model was used for description the viscous fluid drop movement through a bifurcation point. In [26], it was studied a two-phase model of AVM embolization where embolic agent and blood interaction along with its hardening is simulated by viscosity increase.

The above literature review shows the great interest of scientific community to this topic. At the same time, the diversity of approaches and models indi-

At the moment the most common methods of AVM treatment are microsurgical, endovascular and radiosurgical. Endovascular embolization is the selective deactivation of blood vessels from the bloodstream by filling them with a special embolic agent. Modern level of medicine development makes embolization the most preferable method due to minimal invasiveness and possibility of operating in the deep, functionally significant brain areas [7]. This method is widely used. Despite the well-developed embolization technique, the risk of intraoperative vascular rupture remains a concern. According to a study of 408 patients with AVM who were treated endovascular, it was noted 11 % frequency of hemorrhagic complications associated with the treatment [8]. Thus, modeling of AVM embolization is of current interest.

By type of vessels, connecting the arterial and venous basins, AVMs can be divided into fistula (consisting of large vessels) and racemic (consisting of large number of small diameter vessels, which are chaotically intertwined and intersect each other). There are also mixed-type AVMs that combine both racemic and fistula parts. Modern medical examination methods such as CT, MRI and cerebral angiography) do not allow to determine the detailed geometric AVM structure in vivo since its nidus (tangles of abnormal blood vessels) consists of very large number of thin vessels with diameter down to 0.1 mm which exceeds medical equipment resolution. Therefore, various simplified methods for AVM modeling are needed.

Mathematical description of blood flow is often based on one-dimensional approximation of Navier-Stokes equations by averaging over blood vessel cross section. Analytical studies of equations describing flows in pipes with rigid and elastic boundaries can be found in the works of [9], [10], [11]. Viscous fluid flow in a network of soft tubes is modeled in the works of [12], [13], [14] on the base of one-dimensional approximation of mass and momentum conservation laws. Numerical simulation of hemodynamics for large blood vessels based on 3D-1D coupled flow is considered, for example, in the works of [15], [16], [17].

AVM and blood flow in surrounding vessels flow interaction is often studied in analogy with electric and hydraulic networks [18], [19], [20]. Such models make it possible to evaluate the effect of various embolization scenarios on blood flow restructuration and are consistent with the general medical view on AVM hemodynamics. The effect of AVM on the blood flow in the surrounding vessels has also been studied using the mass and momentum conservation laws of the ideal incompressible fluid flow. The resulting equations form a hyperbolic system of differential conservation laws. One dimensional variant of this system on a vessel graph was studied in the works of [21], [22] and its two dimensional variant on branching network channels was considered in the works of [23], [24]. The process of embolization was studied in the work of [25] where two-phase flow model was used for description the viscous fluid drop movement through a bifurcation point. In [26], it was studied a two-phase model of AVM embolization where embolic agent and blood interaction along with its hardening is simulated by viscosity increase.

The above literature review shows the great interest of scientific community to this topic. At the same time, the diversity of approaches and models indi-

cates the presence of a fairly large number of open questions. This is due to the fact that an exhaustive anatomical and physiological description of AVM requires detailed knowledge of malformation nidus structure and mechanical and physiological properties of vessels forming it. Modern diagnostic equipment arsenal does not provide all the necessary information. In this regard used models have sufficiently strong assumptions and simplifications, which however allow to obtain qualitatively correct description of hemodynamics in studied anomaly.

In this paper the flow of blood and embolic agent through the AVM body was considered as filtration flow on the basis of Darcy's law. Such an approach, in our opinion, is justified for small-vascular racemic AVM compartments embolization description. To investigate this process one-dimensional two-phase filtration model is used leading to the special Buckley-Leverett equation in Section 2. This mathematical model represents a scalar conservation law with nonconvex flux for the numerical solution of which we will use a monotonic modification of the second order CABARET scheme [27], see Section 3. This scheme demonstrates higher accuracy in the localization of strong and weak discontinuities compared to other classes of high order shock capturing schemes [28], [29], [30], [31], [32] for which monotonization various nonlinear flux correction procedures are used.

Initial-boundary value AVM problem is considered in Section 4. Numerical modeling of the problem under consideration allows us to describe the evolution of the embolization front, taking into account its partial decay which leads to residual blood content in embolized AVM part, see Section 6. Cardiac cycle effect could be ignored due to modified time used in the problem. Reverse transition to the physical time allows to build a solution corresponding to any given cardiac flow-time dependency. Optimal embolization problem is stated based on this model. Admissible and optimal embolization scenarios, which reduce the risk of rupture of AVM vessels, are considered in Section 7. Mathematically, it is optimal control problem with integral objective function for partial differential equation with state constraints considered in Sections 5, 6. Time dependent boundary condition function is used as control, while constraints arise from medical aspects. The proposed model takes into account the peculiarities of actual embolization process. AVM nidus absolute and relative permeability functions are constructed using intraoperative intracranial measurements in A and B. Mathematical model based on intraoperative data from real patients allowed us to create patient-specific models. This approach can potentially be used in neurosurgical practice to provide a neurosurgeon with recommendation on the course of operation.

cates the presence of a fairly large number of open questions. This is due to the fact that an exhaustive anatomical and physiological description of AVM requires detailed knowledge of malformation nidus structure and mechanical and physiological properties of vessels forming it. Modern diagnostic equipment arsenal does not provide all the necessary information. In this regard used models have sufficiently strong assumptions and simplifications, which however allow to obtain qualitatively correct description of hemodynamics in studied anomaly.

In this paper the flow of blood and embolic agent through the AVM body was considered as filtration flow on the basis of Darcy's law. Such an approach, in our opinion, is justified for small-vascular racemic AVM compartments embolization description. **Some assumptions on embolic agent behaviour, spatial characteristics of AVM, blood rheology and wall properties are made. Specifically, both liquids are assumed to be Newtonian with constant viscosities, incompressible and immiscible. In numerical calculations real blood and embolic agent (ONYX18) viscosities were used. AVM is simplified to one-dimensional case with even distribution of its physical characteristics (porosity, permeability, and cross-sectional area) along model length. Some aspects such as capillary forces, embolic agent adsorption effect, wall properties are neglected.**

To investigate this process one-dimensional two-phase filtration model is used leading to the special Buckley-Leverett equation in Section 2. This mathematical model represents a scalar conservation law with nonconvex flux for the numerical solution of which we will use a monotonic modification of the second order CABARET scheme [27], see Section 3. This scheme demonstrates higher accuracy in the localization of strong and weak discontinuities compared to other classes of high order shock capturing schemes [28], [29], [30], [31], [32] for which monotonization various nonlinear flux correction procedures are used.

Initial-boundary value AVM problem is considered in Section 4. Numerical modeling of the problem under consideration allows us to describe the evolution of the embolization front, taking into account its partial decay which leads to residual blood content in embolized AVM part, see Section 6. Cardiac cycle effect could be ignored due to modified time used in the problem. Reverse transition to the physical time allows to build a solution corresponding to any given cardiac flow-time dependency. Optimal embolization problem is stated based on this model. Admissible and optimal embolization scenarios, which reduce the risk of rupture of AVM vessels, are considered in Section 7. Mathematically, it is optimal control problem with integral objective function for partial differential equation with state constraints considered in Sections 5, 6. Time dependent boundary condition function is used as control, while constraints arise from medical aspects. The proposed model takes into account the peculiarities of actual embolization process. AVM nidus absolute and relative permeability functions are constructed using intraoperative intracranial measurements in A and B. Mathematical model based on intraoperative data from real patients allowed us to create patient-specific models. This approach can potentially be used in neurosurgical practice to provide a neurosurgeon with recommendation on the course of operation.

2 Model description

Racemic AVM compartments with sufficient modeling accuracy can be considered as a porous medium and are the subject of present research. Blood and embolic agent movement through racemic AVM can be modeled as two-phase filtration process, where the displaced phase is blood, and displacing phase is embolic agent. We assume that the structure of the AVM is homogeneous and consider one-dimensional case. It allows us to find out the main patterns of embolization process. In this case the displacement of one phase by another allows a mathematical description first proposed by S. Buckley and M. Leverett [33]. This description is based on the notions of saturation, absolute and relative phase permeabilities [34].

Further the side surface of the AVM is considered impermeable, the cross section, porosity and other parameters of the AVM are assumed to be constant in length and in time, blood and embolic agent are considered as incompressible and immiscible liquids. We neglect the interfacial capillary forces — in this case the pressure in the phases is assumed to be the same. Denote local AVM blood saturation during embolization (blood concentration) as $S(x, t) \in [0, 1]$. Now embolic agent concentration is $1 - S(x, t)$. Under this assumptions the equation for $S(x, t)$ could be written in Buckley-Leverett form [34]:

$$S_t + f(S)_x = 0, \quad (1)$$

where x is the spatial coordinate, t is modified length dimension time:

$$t = \frac{1}{m A} \int Q(\theta) d\theta, \quad t(0) = 0, \quad (2)$$

where θ is physical time, m is AVM nidus porosity, A is AVM cross-section, $Q(\theta)$ is volumetric flow rate of two-phase mixture. Note, that the function Q does not depend on x because of AVM side surface impermeability and phases incompressibility assumptions. In the modified time t the total volumetric flow of two phases and porosity can be considered constant and equal to 1. It allows us to neglect the cardiac cycle effect. Inverse time modification leads to the solution corresponding to time-flow relation.

The saturation function $f(S)$ is the Buckley-Leverett function and is equal to the volume fraction of the displaced liquid (blood) flow in the total flux of the two phases:

$$f(S) = \frac{Q_b}{Q} = \frac{k_b(S)/\eta_b}{k_b(S)/\eta_b + k_e(S)/\eta_e}. \quad (3)$$

Here Q_b is blood volume flow; k_b , k_e are phase relative permeabilities and η_b , η_e are phase dynamic viscosity coefficients, where index b corresponds to blood and index e to embolic agent. Function $f(S)$ increases monotonously from 0 to 1 as blood concentration S grows (figure 1). This function always has a inflection point, which separates concavity and convexity segments. Therefore the Buckley-Leverett model (1), (3) represents the scalar conservation law with a non-convex flux, which allows increasing and decreasing shock waves and also

2 Model description

Racemic AVM compartments with sufficient modeling accuracy can be considered as a porous medium and are the subject of present research. Blood and embolic agent movement through racemic AVM can be modeled as two-phase filtration process, where the displaced phase is blood, and displacing phase is embolic agent. Blood and embolic agent are considered as incompressible and immiscible liquids. We neglect the interfacial capillary forces — in this case the pressure in the phases is assumed to be the same. In one-dimensional case the displacement of one phase by another allows a mathematical description first proposed by S. Buckley and M. Leverett [33]. This description is based on the notions of saturation, absolute and relative phase permeabilities [34]. Denote local AVM blood saturation during embolization (blood concentration) as $S(x, t) \in [0, 1]$. Now embolic agent concentration is $1 - S(x, t)$. Than Buckley-Leverett equation for blood concentration $S(t, x)$ looks as follows:

$$m S_\theta + Q/A f(S)_x = 0, \quad (1)$$

where θ is physical time, x is the spatial coordinate, m is AVM nidus porosity, A is AVM cross-section, Q is volumetric flow rate of two-phase mixture. The saturation function $f(S)$ is the Buckley-Leverett function and is equal to the volume fraction of the displaced liquid (blood) flow in the total flux of the two phases.

We assume that the structure of the AVM is homogeneous. Further the side surface of the AVM is considered impermeable, the cross section, porosity and other parameters of the AVM are assumed to be constant in length and in time. Under this assumptions the equation for $S(t, x)$ could be written in simplified form [34]:

$$S_t + f(S)_x = 0, \quad (2)$$

where t is modified length dimension time:

$$t = \frac{1}{m A} \int Q(\theta) d\theta, \quad t(0) = 0, \quad (3)$$

where θ is physical time, m is AVM nidus porosity, A is AVM cross-section, $Q(\theta)$ is volumetric flow rate of two-phase mixture. Note, that the function Q does not depend on x because of AVM side surface impermeability and phases incompressibility assumptions. In the modified time t the total volumetric flow of two phases and porosity can be considered constant and equal to 1. Various cardiac cycle types could be considered using different forms of function $Q(\theta)$. Inverse time modification leads to the solution of equation (1) corresponding to physical time-flow relation.

The Buckley-Leverett function $f(S)$ mathematically could be presented by formula:

$$f(S) = \frac{Q_b}{Q} = \frac{k_b(S)/\eta_b}{k_b(S)/\eta_b + k_e(S)/\eta_e}. \quad (4)$$

AVM embolization optimal scenario

Figure 1: Typical view of Buckley-Leverett function (3).

rarefaction waves, as well as composite shock-rarefaction waves. The small perturbations in this model propagate with speed $f'(S) \geq 0$, which depends on S in a non-monotonic way and in this paper additionally satisfies the often used conditions:

$$f'(S) > 0 \quad \forall s \in (0, 1), \quad f'(0) = f'(1) = 0. \quad (4)$$

AVM embolization optimal scenario

Figure 1: Typical view of Buckley-Leverett function (4).

Here Q_b is blood volume flow; k_b , k_e are phase relative permeabilities and η_b , η_e are phase dynamic viscosity coefficients, where index b corresponds to blood and index e to embolic agent. Function $f(S)$ increases monotonously from 0 to 1 as blood concentration S grows (figure 1). This function always has an inflection point, which separates concavity and convexity segments. Therefore the Buckley-Leverett model (2), (4) represents the scalar conservation law with a non-convex flux, which allows increasing and decreasing shock waves and also rarefaction waves, as well as composite shock-rarefaction waves. The small perturbations in this model propagate with speed $f'(S) \geq 0$, which depends on S in a non-monotonic way and in this paper additionally satisfies the often used conditions:

$$f'(S) > 0 \quad \forall S \in (0, 1), \quad f'(0) = f'(1) = 0. \quad (5)$$

3 Numerical method

In [35] the second order Upwind Leapfrog scheme was proposed for hyperbolic equations numerical solution. A detailed analysis of this scheme was carried out in the works of [36] and [37], where taking into account the skew-symmetry of its stencil (which is two-point in space and three-layer in time) it was called the CABARET scheme. The main advantages of the CABARET scheme are determined by its compact spatial stencil and for linear transport equation scheme time reversibility and approximation accuracy with two different Courant numbers $r = 0.5$ and $r = 1$, which gives it unique dissipative and dispersive properties [37]. For gas dynamics equations numerical simulation a balance-characteristic version of CABARET scheme was developed by [38]. Taking into account flux variables correction, this scheme showed high accuracy in Blast Wave test calculation [39].

Currently, for numerical simulations of spatially multidimensional gas-dynamic [40] and hydraulic [41] flows, the two-layer in time form of the CABARET scheme [42] is widely used. The monotonicity of this scheme for approximation of scalar conservation law with convex flux was studied in [43] and with a nonconvex flux in [27]. In the present paper, for Buckley-Leverett equation (1), (3) numerical solution the monotonic modification of the CABARET scheme proposed in [27], [44] will be used.

3 Numerical method

In [35] the second order Upwind Leapfrog scheme was proposed for hyperbolic equations numerical solution. A detailed analysis of this scheme was carried out in the works of [36] and [37], where taking into account the skew-symmetry of its stencil (which is two-point in space and three-layer in time) it was called the CABARET scheme. The main advantages of the CABARET scheme are determined by its compact spatial stencil and for linear transport equation scheme time reversibility and approximation accuracy with two different Courant numbers $r = 0.5$ and $r = 1$, which gives it unique dissipative and dispersive properties [37]. For gas dynamics equations numerical simulation a balance-characteristic version of CABARET scheme was developed by [38]. Taking into account flux variables correction, this scheme showed high accuracy in Blast Wave test calculation [39].

Currently, for numerical simulations of spatially multidimensional gas-dynamic [40] and hydraulic [41] flows, the two-layer in time form of the CABARET scheme [42] is widely used. The monotonicity of this scheme for approximation of scalar conservation law with convex flux was studied in [43] and with a nonconvex flux in [27]. In the present paper, for Buckley-Leverett equation (2), (4) numerical solution the monotonic modification of the CABARET scheme proposed in [27], [44] will be used.

4 Initial-boundary value AVM-problem

We denote AVM length by L and total embolization time by T . Initially the AVM is completely filled with blood and does not contain embolic agent. Therefore, the initial data for the Buckley-Leverett equation (1) with flux function (3) is

$$S(0, x) = S_0(x) \equiv 1, \quad x \in [0, L]. \quad (5)$$

The only input of the embolic agent to the AVM is made through the feeding artery, which is set to be the origin $x = 0$. Assuming that the incoming blood concentration is given by the function $g(t)$, we obtain the following boundary condition

$$S(t, 0) = g(t) \in [0, 1], \quad t \in (0, T]. \quad (6)$$

As a result, the equations (1), (3), (5) and (6) set the desired initial-boundary value AVM-problem, which is correct because the characteristics of the equation (1) at the condition $f'(S) > 0$ propagate in the positive direction of the x-axis.

Experience of some neurosurgical operations shows that natural course of the operation is total AVM cross section overlap by embolic agent $g = 0$ followed by its flow reduction in AVM input. Thus, the boundary function $g(t)$ should be as follows

$$g(t) = \begin{cases} 0, & 0 < t < t_1, \\ g^*(t), & t_1 \leq t < t_2, \\ 1, & t_2 \leq t \leq T, \end{cases} \quad (7)$$

where $g^*(t)$ is smooth strictly monotonically increasing function satisfying conditions $g^*(t_1) = 0$ and $g^*(t_2) = 1$.

From the theory of a weak solutions of the hyperbolic conservation laws with a non-convex flux [45], it follows that in the initial time interval $[0, t_1)$ the exact solution of the problem under consideration is an increase shock (vertical segment B_1C_1 on figure 2), to which the centered rarefaction wave (line A_1B_1 on figure 2) is adjacent to the left. This discontinuous solution is given by the formula

$$S(x, t) = \begin{cases} a^{-1}(x/t), & 0 \leq x < Dt, \\ 1, & Dt < x \leq L, \end{cases} \quad (8)$$

where a^{-1} is function, inverse to function f' ;

$$D = \frac{1 - f(S_1)}{1 - S_1} \quad (9)$$

is the shock velocity, which is determined from the Hugoniot condition for the conservation law (1); S_1 is the horizontal coordinate of the point B in figure 1 in which the straight line passing through the point $C = (1, 1)$ touches the graph of the function $f(S)$; the value of S_1 is calculated from the equation

$$f(S_1) + f'(S_1)(1 - S_1) = 1. \quad (10)$$

Solid line on figure 2 shows exact solution (8)-(9) for Buckley-Leverett function $f(S) = \frac{S^3/4}{(1-S)^2/9 + S^3/4}$ at time $t = 3.12 < t_1$ while circles present numerical

4 Initial-boundary value AVM-problem

We denote AVM length by L and total embolization time by T . Initially the AVM is completely filled with blood and does not contain embolic agent. Therefore, the initial data for the Buckley-Leverett equation (2) with flux function (4) is

$$S(0, x) = S_0(x) \equiv 1, \quad x \in [0, L]. \quad (6)$$

The only input of the embolic agent and blood mixture to the AVM is made through the feeding artery, which is set to be the origin $x = 0$. Blood concentration here is given by the function $g(t)$, determined by operation strategy. Thus we obtain the following boundary condition:

$$S(t, 0) = g(t) \in [0, 1], \quad t \in (0, T]. \quad (7)$$

As a result, the equations (2), (4), (6) and (7) set the desired initial-boundary value AVM-problem. It is correct because the characteristics of the equation (2) propagate in the positive direction of the x-axis while $f'(S) > 0$ which is true for $S \in (0, 1)$.

Experience of some neurosurgical operations shows that natural course of the operation is total AVM cross section overlap by embolic agent (which mathematically is $g = 0$) followed by embolic agent concentration reduction (blood concentration increase) in AVM input. Thus, the boundary function $g(t)$ could be chosen as follows:

$$g(t) = \begin{cases} 0, & 0 < t < t_1, \\ g^*(t), & t_1 \leq t < t_2, \\ 1, & t_2 \leq t \leq T, \end{cases} \quad (8)$$

where $g^*(t)$ is smooth strictly monotonically increasing function satisfying conditions $g^*(t_1) = 0$ and $g^*(t_2) = 1$.

From the theory of a weak solutions of the hyperbolic conservation laws with a non-convex flux [45], it follows that in the initial time interval $[0, t_1)$ the exact solution of the problem under consideration is an increase shock (vertical segment B_1C_1 on figure 2), to which the centered rarefaction wave (line A_1B_1 on figure 2) is adjacent to the left. This discontinuous solution is given by the formula

$$S(x, t) = \begin{cases} a^{-1}(x/t), & 0 \leq x < Dt, \\ 1, & Dt < x \leq L, \end{cases} \quad (9)$$

where a^{-1} is function, inverse to function f' ;

$$D = \frac{1 - f(S_1)}{1 - S_1} \quad (10)$$

is the shock velocity, which is determined from the Hugoniot condition for the conservation law (2); S_1 is the horizontal coordinate of the point B in figure 1 in which the straight line passing through the point $C = (1, 1)$ touches the graph of the function $f(S)$; the value of S_1 is calculated from the equation

$$f(S_1) + f'(S_1)(1 - S_1) = 1. \quad (11)$$

AVM embolization optimal scenario

Figure 2: Exact (line) and numerical solution (circles) comparison for Buckley-Leverett function $f(S) = \frac{S^{3/4}}{(1-S)^3/9+S^{3/4}}$ with $t < t_1$.

solution obtained by CABARET scheme. One can see a good concurrence between numerical and exact solutions both at discontinuity and in rarefaction wave region. Such comparison of exact and numerical solutions for $t < t_1$ was used to control the accuracy of calculations with various clinically based Buckley-Leverett functions (Appendix B).

On the time interval $[t_1, T]$, the exact solution of the problem (1), (3), (5) and (6) becomes more complex and for its calculation we will use the CABARET scheme. A significant advantage of the CABARET scheme compared to WENO type schemes [27] is that this scheme is defined on a compact three-point spatial stencil located inside a single cell of the difference grid, and therefore, when calculating initial boundary-value problems, it is not necessary for the CABARET scheme to apply auxiliary asymmetric difference equations in the near-boundary grid nodes.

To solve optimal embolization problem Buckley-Leverett functions (3) were constructed for several patients using clinical data of hemodynamic parameters from neurosurgical operations at National Medical Research Center named after academic E. N. Meshalkin [46]. Clinically based Buckley-Leverett function construction are given in the Appendix B.

AVM embolization optimal scenario

Figure 2: Exact (line) and numerical solution (circles) comparison for Buckley-Leverett function $f(S) = \frac{S^{3/4}}{(1-S)^3/9+S^{3/4}}$ with $t < t_1$.

Solid line on figure 2 shows exact solution (9)-(10) for Buckley-Leverett function $f(S) = \frac{S^{3/4}}{(1-S)^3/9+S^{3/4}}$ at time $t = 3.12 < t_1$ while circles present numerical solution obtained by CABARET scheme. One can see a good concurrence between numerical and exact solutions both at discontinuity and in rarefaction wave region. Such comparison of exact and numerical solutions for $t < t_1$ was used to control the accuracy of calculations with various clinically based Buckley-Leverett functions (Appendix B).

On the time interval $[t_1, T]$, the exact solution of the problem (2), (4), (6) and (7) becomes more complex and for its calculation we will use the CABARET scheme. A significant advantage of the CABARET scheme compared to WENO type schemes [27] is that this scheme is defined on a compact three-point spatial stencil located inside a single cell of the difference grid, and therefore, when calculating initial boundary-value problems, it is not necessary for the CABARET scheme to apply auxiliary asymmetric difference equations in the near-boundary grid nodes.

To solve optimal embolization problem Buckley-Leverett functions (4) were constructed for several patients using clinical data of hemodynamic parameters from neurosurgical operations at National Medical Research Center named after academic E. N. Meshalkin [46]. Clinically based Buckley-Leverett function construction are given in the Appendix B.

5 Unit load as restriction factor

The choice of the value characterizing the risk of AVM rupture is an important question arising in the study of embolization process. To answer this question the notion of unit load was introduced [47]. Unit load is the mean value of energy released per unit of time with blood passing through the unit of AVM volume. It is given by formula: $(E_1 - E_2)/V$, where $E_1(t)$ is the energy of the incoming blood flow, $E_2(t)$ is the energy of the outgoing blood flow, $V(t)$ is the AVM volume part occupied by blood.

AVM embolization degree $D(t)$ is another important parameter and is defined as the fraction of AVM vascular space occupied by embolic agent. It can vary from 0 to 1 and for the considered embolization model can be calculated by formula:

$$D(t) = 1 - \frac{1}{L} \int_0^L S(t, x) dx. \quad (11)$$

Unit load behavior for real patients using clinical data was studied in the work of [47]. The connection between unit load and AVM embolization degree was established. It was noted that unit load begins to increase sharply with embolization degree of more than 60%. Two patient groups were studied in National Medical Research Center named after academic E. N. Meshalkin. In the first group the unit load sharp increase near 60% embolization degree was taken into account and operations ended earlier than this value was achieved. In the second group the operations were carried out according without taking into account the unit load. It was shown that accounting for the unit load reduced the risk of complication (hemorrhage). It should be noted that embolization degree can differ from 60% and is determined individually by neurosurgeon for each patient, based on the patient's condition. For example, for small malformations with one supply vessel, a complete embolization is usually performed in one stage. Further in the paper without loss of generality the case of the 60% maximum admissible embolization degree is considered.

It is known that simultaneous exclusion of high flow AVMs is accompanied by elevated risk of unfavorable bloodstream redistribution and, as a result, perioperative hemorrhage. In this regard, there is a practice of embolization dividing into stages. In current study embolization degree is used as the value characterizing the risk of AVM rupture. On the one hand, value of embolization degree greater than 60% is considered dangerous. On the other hand, too small values of embolization degree show that operation was not effective since the pathological vascular bed remains widely open to blood flow and the number of surgical interventions unjustifiably increase. Thus, it is necessary to find some compromise between the risk of complications and the most complete exclusion of the AVM from the bloodstream per one stage.

5 Unit load as restriction factor

The choice of the value characterizing the risk of AVM rupture is an important question arising in the study of embolization process. To answer this question the notion of unit load was introduced [47]. Unit load is the mean value of energy released per unit of time with blood passing through the unit of AVM volume. It is given by formula: $(E_{in} - E_{out})/V$, with $E_i = Q_i(t)P_i(t)$, where E_i are blood energy flows on AVM nidus input and output, Q_i, P_i are volumetric blood flow and total blood pressure with $i \in (in, out)$. $V(t)$ is the AVM volume part occupied by blood.

AVM embolization degree $D(t)$ is another important parameter and is defined as the fraction of AVM vascular space occupied by embolic agent. It can vary from 0 to 1 and for the considered embolization model can be calculated by formula:

$$D(t) = 1 - \frac{1}{L} \int_0^L S(t, x) dx. \quad (12)$$

Unit load behavior for real patients using clinical data was studied in the work of [47]. The connection between unit load and AVM embolization degree was established. It was noted that unit load begins to increase sharply with embolization degree of more than 60%. Two patient groups were studied in National Medical Research Center named after academic E. N. Meshalkin. In the first group the unit load sharp increase near 60% embolization degree was taken into account and operations ended earlier than this value was achieved. In the second group the operations were carried out according without taking into account the unit load. It was shown that accounting for the unit load reduced the risk of complication (hemorrhage). It should be noted that embolization degree can differ from 60% and is determined individually by neurosurgeon for each patient, based on the patient's condition. For example, for small malformations with one supply vessel, a complete embolization is usually performed in one stage. Further in the paper without loss of generality the case of the 60% maximum admissible embolization degree is considered.

It is known that simultaneous exclusion of high flow AVMs is accompanied by elevated risk of unfavorable bloodstream redistribution and, as a result, perioperative hemorrhage. In this regard, there is a practice of embolization dividing into stages. In current study embolization degree is used as the value characterizing the risk of AVM rupture. On the one hand, value of embolization degree greater than 60% is considered dangerous. On the other hand, too small values of embolization degree show that operation was not effective since the pathological vascular bed remains widely open to blood flow and the number of surgical interventions unjustifiably increase. Thus, it is necessary to find some compromise between the risk of complications and the most complete exclusion of the AVM from the bloodstream per one stage.

6 Blood conductivity as embolization success factor

Along with embolization degree which characterizes AVM fullness by embolic agent, its distribution inside AVM is important. Residual blood flow through the AVM is determined by how much we can reduce AVM blood conductivity, which directly depends on embolic agent distribution in AVM nidus. Blood conductivity can be determined by Darcy's differential law for blood:

$$Q_b = -K A \frac{k_b(S)}{\eta_b} \frac{\partial p}{\partial x}. \quad (12)$$

Here $p(t, x)$ is pressure distribution and $Q_b(t, x)$ is blood volume flow distribution in the AVM. Thus AVM local resistance to advancement of a blood flow is $R(t, x) = \eta_b / (K A k_b(S))$. Then AVM full blood conductivity is given by formula

$$\hat{G}(t) = \left(\int_0^L R(t, x) dx \right)^{-1}. \quad (13)$$

To determine the residual blood flow it is convenient to use dimensionless quantities. Let us define relative blood conductivity $G(t) = \hat{G}(t)/\hat{G}(0)$, $G(t) \in (0, 1]$ and name it blood conductivity degree. Numerically blood conductivity degree is calculated using formula

$$G(t) = \left(\frac{1}{N} \sum_{i=1}^N \frac{1}{k_b(S_i)} \right)^{-1}, \quad (14)$$

where N is the number of intervals of a uniform partition of the segment $[0, L]$, S_i is the calculated blood concentration value on i th interval. If on some interval k_b is a zero value, then conductivity $G(t)$ is considered equal to zero.

The reduction of AVM section available for blood flow (increase in degree of AVM section cut off by embolic agent) reduces blood conductivity degree and, therefore, has a positive effect on embolization results. Further it is illustrated with two examples calculated for patient P Buckley-Leverett function (figure 9). Clinically based AVM geometry parameters are given in the Appendix A and Buckley-Leverett function construction are given in the Appendix B.

First example on the figure 3 shows bad embolization scenario. The boundary conditions (7) are: $g^*(t) = (t - 5)/5$, $t_1 = 5$, $t_2 = 10$. It means that only embolic agent enters the AVM input at $t < t_1$. Despite this fact on the embolization front it occupies only part of the cross-section. This is due to the decay of the discontinuity formed at the initial moment, which is unstable according to Oleinik-Liu criterion [45]. Further, at $t \geq t_1$ embolic agent concentration near AVM input starts to decrease and the second discontinuity is formed at $t = t_1$. Embolic agent concentration on embolization front $1 - S \approx 0.15$ remains constant until $t = 13$. Since at this moment posterior discontinuity reaches the

6 Blood conductivity as embolization success factor

Along with embolization degree which characterizes AVM fullness by embolic agent, its distribution inside AVM is important. Residual blood flow through the AVM is determined by how much we can reduce AVM blood conductivity, which directly depends on embolic agent distribution in AVM nidus. Blood conductivity can be determined by Darcy's differential law for blood:

$$Q_b = -K A \frac{k_b(S)}{\eta_b} \frac{\partial p}{\partial x}. \quad (13)$$

Here $p(t, x)$ is pressure distribution and $Q_b(t, x)$ is blood volume flow distribution in the AVM. Thus AVM local resistance to advancement of a blood flow is $R(t, x) = \eta_b / (K A k_b(S))$. Then AVM full blood conductivity is given by formula

$$\hat{G}(t) = \left(\int_0^L R(t, x) dx \right)^{-1}. \quad (14)$$

To determine the residual blood flow it is convenient to use dimensionless quantities. Let us define relative blood conductivity $G(t) = \hat{G}(t)/\hat{G}(0)$, $G(t) \in (0, 1]$ and name it blood conductivity degree. Numerically blood conductivity degree is calculated using formula

$$G(t) = \left(\frac{1}{N} \sum_{i=1}^N \frac{1}{k_b(S_i)} \right)^{-1}, \quad (15)$$

where N is the number of intervals of a uniform partition of the segment $[0, L]$, S_i is the calculated blood concentration value on i th interval. If on some interval k_b is a zero value, then conductivity $G(t)$ is considered equal to zero. Note, that any thin region that could be completely occluded will not be stable not practically nor theoretically (in the model) and breaks up starting from the leading edge due to Oleinik-Liu condition [45].

The reduction of AVM section available for blood flow (increase in degree of AVM section cut off by embolic agent) reduces blood conductivity degree and, therefore, has a positive effect on embolization results. Further it is illustrated with two examples calculated for patient P Buckley-Leverett function (figure 9). Clinically based AVM geometry parameters are given in the Appendix A and Buckley-Leverett function construction are given in the Appendix B.

First example on the figure 3 shows bad embolization scenario. The boundary conditions (8) are: $g^*(t) = (t - 5)/5$, $t_1 = 5$, $t_2 = 10$. It means that only embolic agent enters the AVM input at $t < t_1$. Despite this fact on the embolization front it occupies only part of the cross-section. This is due to the decay of the discontinuity formed at the initial moment, which is unstable according to Oleinik-Liu criterion. Further, at $t \geq t_1$ embolic agent concentration

AVM embolization optimal scenario

anterior one (figure 3(b)), AVM section cut off by embolic agent begins to decrease monotonically (figure 3(c)) leading to a significant increase in AVM blood conductivity. In this case after the end of operation the blood flow through the AVM is largely maintained.

Figure 4 shows successful (good) embolization scenario. The boundary conditions (7) are: $g^*(t) = (t - 5)/30$, $t_1 = 5$, $t_2 = 35$. Here the operation start is the same as in the previous example, but for $t \geq t_1$ the incoming embolic agent concentration decreases slower than previously. In this case the posterior discontinuity moves not fast enough to catch up with the anterior one and the embolic agent concentration on embolization front does not decrease. AVM cross section cut off will always be not less than on embolization front. In this case blood conductivity grows much slower over time. Blood conductivity dynamics for both scenarios are presented on figure 5.

This cases demonstrate that operation success depends on embolic agent income $g(t)$, which determines whether AVM cross section will be cut off by embolic agent the best possible way, or over time a part of AVM section previously blocked by the embolic agent will be reopened for blood flow and blood conductivity will grow.

AVM embolization optimal scenario

near AVM input starts to decrease and the second discontinuity is formed at $t = t_1$. Embolic agent concentration on embolization front $1 - S \approx 0.15$ remains constant until $t = 13$. Since at this moment posterior discontinuity reaches the anterior one (figure 3(b)), AVM section cut off by embolic agent begins to decrease monotonically (figure 3(c)) leading to a significant increase in AVM blood conductivity. In this case after the end of operation the blood flow through the AVM is largely maintained.

Figure 4 shows successful (good) embolization scenario. The boundary conditions (8) are: $g^*(t) = (t - 5)/30$, $t_1 = 5$, $t_2 = 35$. Here the operation start is the same as in the previous example, but for $t \geq t_1$ the input embolic agent concentration decreases slower than previously. In this case the posterior discontinuity moves not fast enough to catch up with the anterior one and the embolic agent concentration on embolization front does not decrease. AVM cross section cut off will always be not less than on embolization front. In this case blood conductivity grows much slower over time. Blood conductivity dynamics for both scenarios are presented on figure 5.

This cases demonstrate that operation success depends on embolic agent input $g(t)$, which determines whether AVM cross section will be cut off by embolic agent the best possible way, or over time a part of AVM section previously blocked by the embolic agent will be reopened for blood flow and blood conductivity will grow.

AVM embolization optimal scenario

Figure 5: Comparison of blood conductivity G for patient P: good (lower curve) and bad (upper curve) embolization scenarios. The dashed line indicates the time when two discontinuities merge in a bad scenario.

7 Optimal embolization problem

Given the above, the following formulation of the optimal embolization problem as an optimal control problem is proposed:

To minimize the linear combination of AVM blood conductivity degree and AVM fullness by blood at terminal time using embolic agent input as control.

The following conditions must be fulfilled:

- (1) Embolization degree should not exceed 60%.
- (2) For medical reasons embolic agent should not reach the AVM exit, that is, it should not enter the vein until the complete blockage of the AVM nidus.
- (3) The operation is considered completed at time T when embolic agent ceases to enter the AVM input. In this regard, when modeling only the time interval $[0, T]$ is considered.

In other words, it is necessary to choose a boundary control $g(t)$ such that the solution of the problem (1), (5), (6) delivers minimum to the functional:

$$J = \alpha(1 - D(T)) + (1 - \alpha)G(T), \quad \alpha \in (0, 1), \quad (15)$$

and satisfies the following conditions:

$$D(t) \leq 0.6, \quad t \in [0, T]; \quad (16)$$

$$S(t, L) = 1, \quad t \in [0, T]; \quad (17)$$

$$S(t, 0) < 1, \quad t \in (0, T). \quad (18)$$

Problem is solved for fixed parameter α . Further this problem for special embolization mode will be considered for specific patients and the results will show

AVM embolization optimal scenario

Figure 5: Comparison of blood conductivity G for patient P: good (lower curve) and bad (upper curve) embolization scenarios. The dashed line indicates the time when two discontinuities merge in a bad scenario.

7 Optimal embolization problem

Given the above, the following formulation of the optimal embolization problem as an optimal control problem is proposed:

To minimize the linear combination of AVM blood conductivity degree and AVM fullness by blood at terminal time using embolic agent input as control.

The following conditions must be fulfilled:

- (1) Embolization degree should not exceed 60%.
- (2) For medical reasons embolic agent should not reach the AVM exit, that is, it should not enter the vein until the complete blockage of the AVM nidus.
- (3) The operation is considered completed at time T when embolic agent ceases to enter the AVM input. In this regard, when modeling only the time interval $[0, T]$ is considered.

In other words, it is necessary to choose a boundary control $g(t)$ such that the solution of the problem (2), (6), (7) delivers minimum to the functional:

$$J = \alpha(1 - D(T)) + (1 - \alpha)G(T), \quad \alpha \in (0, 1), \quad (16)$$

and satisfies the following conditions:

$$D(t) \leq 0.6, \quad t \in [0, T]; \quad (17)$$

$$S(t, L) = 1, \quad t \in [0, T]; \quad (18)$$

$$S(t, 0) < 1, \quad t \in (0, T). \quad (19)$$

Problem is solved for fixed parameter α . Further this problem for special embolization mode will be considered for specific patients and the results will show

that optimal control problem solution delivers minimum for functional for all $\alpha \in (0, 1)$ simultaneously.

7.1 Optimal scenario for special embolization case

Let us consider problem (1), (5), (6) solution for clinically based AVM geometry and Buckley-Leverett functions $f(S)$ (these parameters are given in the Appendix A, Appendix B) and piecewise-linear function $g(t)$ (7) of special type:

$$g^*(t) = (t - t_1)/t_*, \quad t_2 = t_1 + t_*, \quad t_1 > 0, \quad t_* > 0. \quad (19)$$

This function approximates the natural embolization sequence: full AVM cross section cut off at the beginning with decreasing embolic agent to the end of operation. Thus, in these scenarios embolic agent input is controlled by two parameters: t_1 and t_* , with end embolization time $T = t_1 + t_*$. Admissible control class (7) with $g^*(t)$ from (19) is extended by $t_* = 0$, meaning that function $g(t)$ has the discontinuity of first type at $t = t_1$:

$$g(t) = \begin{cases} 0 & , \quad 0 < t < t_1 \\ 1 & , \quad t_1 \leq t \leq T \end{cases}, \quad t_* = 0 \quad (20)$$

To solve optimal embolization problem for the extended admissible control class (19)-(20), problem (1), (5), (6) calculations were made for different values of parameters t_1 and t_* . Values of t_1 vary from 1 to 30 with step 1 while t_* varies from 0.1 to 40.1 with step 1.

The calculations were carried out on a rectangular grid. The spatial coordinate x changes from 0 to L with constant step h . Number of space partition points is 200. Note, that choice of L does not limit problem generality because it can be set any given value by change of variables in (1). Number of time partition points was 2500 at maximum, and time step is given by

$$\tau = rh/B, \quad B = \max_{S \in [0,1]} f'(S), \quad (21)$$

where the Courant number is $r = 0.5$, and Buckley-Leverett function derivative maximum $f'(S)$ is determined individually for each patient. The number of time steps was chosen in such a way that for each patient for all embolization scenarios the embolic agent reached the vein. Further calculations have no practical meaning.

It is convenient to switch to (t_1, T) plane to study optimal embolization problem. On this plane only final states of embolization process are considered. In this case straight lines parallel to coordinate angle bisector are lines of constant values of t_* . The bisector itself stays for $t_* = 0$ and the area under it has no sense. Since embolic agent does not exit the malformation then condition (16) fulfillment at terminal time T means that it is valid for all times $[0, T]$.

Let us analyze optimal embolization problem (15)-(18) solution construction on the example of patients whose data is shown on figure 9. For this case plane (t_1, T) is presented on figure 6. On these figures fill color determines blood

that optimal control problem solution delivers minimum for functional for all $\alpha \in (0, 1)$ simultaneously.

7.1 Optimal scenario for special embolization case

Let us consider problem (2), (6), (7) solution for clinically based AVM geometry and Buckley-Leverett functions $f(S)$ (these parameters are given in the Appendix A, Appendix B) and boundary condition of special type given by piecewise-linear function:

$$g(t) = \begin{cases} 0, & 0 < t < t_1, \\ (t - t_1)/t_*, & t_1 \leq t < t_2, \\ 1, & t_2 \leq t \leq T, \end{cases} \quad (20)$$

where $t_* = t_2 - t_1$, $t_1 > 0$, $t_2 > t_1$.

This function approximates the natural embolization process: full AVM cross section cut off at the beginning with decreasing embolic agent to the end of operation. Thus, in these scenarios embolic agent input is controlled by two parameters: t_1 and t_* , with end embolization time $T = t_1 + t_*$. Admissible control class (20) is extended by $t_* = 0$, meaning that function $g(t)$ has the discontinuity of first type at $t = t_1$:

$$g(t) = \begin{cases} 0 & , \quad 0 < t < t_1 \\ 1 & , \quad t_1 \leq t \leq T \end{cases}, \quad t_* = 0 \quad (21)$$

To solve optimal embolization problem for the extended admissible control class (20)-(21), problem (2), (6), (7) calculations were made for different values of parameters t_1 and t_* . Values of t_1 vary from 1 to 30 with step 1 while t_* varies from 0.1 to 40.1 with step 1.

The calculations were carried out on a rectangular grid. The spatial coordinate x changes from 0 to L with constant step h . Number of space partition points is 200. Note, that choice of L does not limit problem generality because it can be set any given value by change of variables in (2). Number of time partition points was 2500 at maximum, and time step is given by

$$\tau = rh/B, \quad B = \max_{S \in [0,1]} f'(S), \quad (22)$$

where the Courant number is $r = 0.5$, and Buckley-Leverett function derivative maximum $f'(S)$ is determined individually for each patient. The number of time steps was chosen in such a way that for each patient for all embolization scenarios the embolic agent reached the vein. Further calculations have no practical meaning. Moreover, embolic agent does not exit the malformation then condition (17) fulfillment at terminal time T means that it is valid for all times $[0, T]$.

It is convenient to switch to (t_1, T) parameters plane to study optimal embolization problem (see figure 6). On this plane only final states of embolization process are considered. In this case straight lines parallel to coordinate angle

AVM embolization optimal scenario

conductivity degree $G(T)$ with level lines shown in solid lines. Dashed lines are level lines of embolization degree $D(T)$ with 60% level shown in bold solid line. Points on the rights of this line stay for embolization degree greater than 60%. Blue area shows regimes with embolic agent reaching the vein. A small neighborhood of the origin should be excluded due to obvious medical condition $T > 0$. Thus for this case admissible parameters area is A_1ABCD . Let us show that for considered patients point B , which is the intersection of coordinate angle bisector and 60% embolization degree level line, corresponds to the optimal embolization regime.

Let us minimize objective functional (15) using gradient descent method for $\alpha \in (0, 1)$. Objective functional antigradient $-\nabla J = \alpha \nabla D - (1 - \alpha) \nabla G$ is a convex linear combination of blood conductivity degree G and function $1 - D$ antigradients. Figure 6(b) shows that for all $\alpha \in (0, 1)$ antigradient $-\nabla J$ lays between ∇D and $-\nabla G$ in sector Ω . Location of blood conductivity degree G and embolization degree D level lines on figure 6 illustrates that movement from any interior point E of admissible parameters domain along the objective functional antigradient leads us to the boundary of embolization admissible parameters area A_1ABCD since in a bounded area of admissible parameters the objective functional has no internal extrema. It should be noted that starting from the part of the boundary A_1D , we immediately go inside the region of admissible parameters, since objective functional decreases with fixed T while parameter t_1 grows. Movement along the part of boundary $\Gamma = A_1ABCD \setminus A_1D$ towards functional descending leads us to point B , which delivers objective functional minimum. Note, that ∇G on boundary part AB and ∇D on boundary part BC projections are empty, since AB and BC are G and D level lines respectively. But since $\alpha \in (0, 1)$ then objective functional gradient projection on the boundary is nonempty on Γ .

The found optimal embolization regime, given by point B , is discontinuous and corresponds to the function (20). At $t = 0$ there is no embolic agent in AVM inlet, at $t \in (0, T)$ AVM inlet is completely blocked by embolic agent, and at terminal time $t = T$ embolization is completed and there is no more embolic agent in AVM inlet. At the end of operation time T embolization degree is 60%. Figure 7 shows corresponding solution for patient P in two consecutive times.

Figure 6 shows that for all patients of concern following simple qualitative conditions are valid:

- (1) inside the area A_1ABCD objective functional decreases with fixed T while parameter t_1 grows and has no internal minima;
- (2) functional decreases on the boundary while moving along its parts A_1AB and DCB towards point B ;
- (3) embolic agent does not reach the vein during operation with parameters values corresponding to point B .

It is easy to prove that these conditions are sufficient for optimal embolization regime to correspond to the same point B .

AVM embolization optimal scenario

bisector are lines of constant values of $t_* = T - t_1$. The bisector line itself stays for parameter $t_* = 0$ and the area under this line is not considered because of $T < t_1$, which means embolization is finished before time t_1 is reached.

Let us analyze optimal embolization problem (16)–(19) solution construction on the example of patients whose data is shown on figure 9. For this case planes (t_1, T) is presented on figure 6. On these figures fill color determines blood conductivity degree $G(T)$ with level lines shown in solid lines (wiggling is caused by mesh data interpolation). Dashed lines are level lines of embolization degree $D(T)$ with 60% level shown in bold solid line. Points on the rights of this line stay for embolization degree greater than 60%. Purple area shows regimes with embolic agent reaching the vein, where condition (18) is violated. Since operation duration T is positive (see (7)), than a small neighborhood of the origin should be excluded. Thus for this case admissible parameters area is A_1ABCD . Let us show that for considered patients point B , which is the intersection of coordinate angle bisector and 60% embolization degree level line, corresponds to the optimal embolization regime.

Let us minimize objective functional (16) using gradient descent method for $\alpha \in (0, 1)$. Objective functional antigradient $-\nabla J = \alpha \nabla D - (1 - \alpha) \nabla G$ is a convex linear combination of blood conductivity degree G and function $1 - D$ antigradients. Figure 6(b) shows that for all $\alpha \in (0, 1)$ antigradient $-\nabla J$ lays between ∇D and $-\nabla G$ in sector Ω . Location of blood conductivity degree G and embolization degree D level lines on figure 6 illustrates that movement from any interior point E of admissible parameters domain along the objective functional antigradient leads us to the boundary of embolization admissible parameters area A_1ABCD since in a bounded area of admissible parameters the objective functional has no internal extrema. It should be noted that starting from the part of the boundary A_1D , we immediately go inside the region of admissible parameters, since objective functional decreases with fixed T while parameter t_1 grows. Movement along the part of boundary $\Gamma = A_1ABCD \setminus A_1D$ towards functional descending leads us to point B , which delivers objective functional minimum. Note, that ∇G on boundary part AB and ∇D on boundary part BC projections are empty, since AB and BC are G and D level lines respectively. But since $\alpha \in (0, 1)$ then objective functional gradient projection on the boundary is nonempty on Γ .

The found optimal embolization regime, given by point B , is discontinuous and corresponds to the function (21). At $t = 0$ there is no embolic agent in AVM inlet, at $t \in (0, T)$ AVM inlet is completely blocked by embolic agent, and at terminal time $t = T$ embolization is completed and there is no more embolic agent in AVM inlet. At the end of operation time T embolization degree is 60%. Figure 7 shows corresponding solution for patient P in two consecutive times.

Figure 6 shows that for all patients of concern following simple qualitative conditions are valid:

- (1) inside the area A_1ABCD objective functional decreases with fixed T while parameter t_1 grows and has no internal minima;
- (2) functional decreases on the boundary while moving along its parts A_1AB

AVM embolization optimal scenario

The course of embolization is determined by AVM length L , Buckley-Leverett function $f(S)$, which describes malformation properties, and control $g(t)$, which sets embolization regime. Qualitative character of conditions (1)–(3) entail that for Buckley-Leverett functions close to considered embolization regimes will be similar. In particular, it means that Buckley-Leverett function construction from clinical data could be made with sufficiently big error. This conditions will also be valid for close to considered piecewise-linear and nonlinear classes of control function $g(t)$. For all this classes optimal embolization regime will also lay on the intersection of the main diagonal and level line of maximal admissible embolization degree.

Generally speaking, the situation when embolic agent reaches the vein earlier then 60% embolization degree is achieved is not excluded. In this case boundary A_1ABCD part BC disappears and optimal regime lays on the intersection of blue area boundary and main diagonal and also is discontinuous. This case did not appear among the considered patients.

Practically, it is impossible to maintain optimal regime exactly. But since objective functional value varies insignificantly in the neighborhood of point B then regime from the neighborhood of point B could be quite acceptable.

AVM embolization optimal scenario

and DCB towards point B ;

- (3) embolic agent does not reach the vein during operation with parameters values corresponding to point B .

It is easy to prove that these conditions are sufficient for optimal embolization regime to correspond to the same point B .

The course of embolization is determined by AVM length L , Buckley-Leverett function $f(S)$, which describes malformation properties, and control $g(t)$, which sets embolization regime. Qualitative character of conditions (1)–(3) entail that for Buckley-Leverett functions close to considered embolization regimes will be similar. In particular, it means that Buckley-Leverett function construction from clinical data could be made with sufficiently big error. This conditions will also be valid for close to considered piecewise-linear and nonlinear classes of control function $g(t)$. For all this classes optimal embolization regime will also lay on the intersection of the main diagonal and level line of maximal admissible embolization degree.

Generally speaking, the situation when embolic agent reaches the vein earlier then 60% embolization degree is achieved is not excluded. In this case boundary A_1ABCD part BC disappears and optimal regime lays on the intersection of purple area boundary and main diagonal and also is discontinuous. This case did not appear among the considered patients.

Practically, it is impossible to maintain optimal regime exactly. But since objective functional value varies insignificantly in the neighborhood of point B then regime from the neighborhood of point B could be quite acceptable.

8 Results and discussion

In this paper optimal AVM embolization scenario in terms of safety and effectiveness was studied. Joint blood and embolic agent flow in AVM nidus was described using one dimensional two-phase filtration model. For simulation Godunov's monotonic modification of CABARET scheme was used, which allows calculation of discontinuous solutions for scalar conservation law with a non-convex flow. Absolute permeabilities for intact malformations were determined for real patients in vivo. Also, Buckley-Leverett functions for malformations were constructed based on the clinical data of real patients, and were used to study the optimal embolization problem. AVM optimal embolization problem was formulated, its numerical solution for special linear embolization regime was constructed, and admissible and optimal embolization scenarios were calculated. The optimal scenario for all examined clinical cases corresponds to a discontinuous embolization regime. Namely, complete closure of AVM inlet cross section with an embolic agent, then bringing the embolic agent amount in the AVM nidus to admissible maximum and a sharp cessation of embolic agent delivery.

Available methods for AVM geometry reconstruction from neuroimaging data (CT, MRI, cerebral angiography) make it possible to distinguish vessels with an average diameter of at least 0.5 mm. It is insufficient to determine in vivo the detailed geometric AVM structure, which consists of a very large number of intertwined thin vessels whose diameter can reach up to 0.1 mm. Therefore, when modeling AVMs, it becomes necessary to use simplified approaches. The paper considers a one-dimensional model of AVM embolization, which allows us to describe important qualitative patterns of this process. In addition to the difficulty of AVM geometry reconstruction, the mathematical description of blood rheology, arterial and venous vessels walls properties, the forces arising in vessels walls, as well as the forces acting on the vessel from the surrounding tissues, the filtration processes through the vessel wall, chemical reactions, and other ongoing processes are still far away to completion. Despite this, taking into account the most significant and determining parameters of the system allows one to build mathematical models that give a qualitatively correct description of its behavior.

For a more detailed study of embolization in future, it is necessary to consider more accurate models that take into account additional characteristics of the AVM embolization process. This primarily concerns the uneven distribution of the physical characteristics of the AVM (porosity, permeability, and cross-sectional area) along model length. More accurate embolic agent behavior description is also important since some of its varieties are liquids with pronounced non-Newtonian properties. In addition, embolic agent adsorption in blood vessels walls and its properties variation over time during solidification are of interest. Despite the impossibility at present of complete AVM internal structure restoration using neuroimaging data, macroscopic characteristics such as the shape of cross sections and the length of AVM nidus can be restored with sufficient for modeling accuracy. In this regard, the next stage of the study may

8 Discussion

In this paper optimal AVM embolization scenario in terms of safety and effectiveness was studied. Joint blood and embolic agent flow in AVM nidus was described using one dimensional two-phase filtration model. For simulation Godunov's monotonic modification of CABARET scheme was used, which allows calculation of discontinuous solutions for scalar conservation law with a non-convex flow. Absolute permeabilities for intact malformations were determined for real patients in vivo. Also, Buckley-Leverett functions for malformations were constructed based on the clinical data of real patients, and were used to study the optimal embolization problem. AVM optimal embolization problem was formulated, its numerical solution for special linear embolization regime was constructed, and admissible and optimal embolization scenarios were calculated. The optimal scenario for all examined clinical cases corresponds to a discontinuous embolization regime. Namely, complete closure of AVM inlet cross section with an embolic agent, then bringing the embolic agent amount in the AVM nidus to admissible maximum and a sharp cessation of embolic agent delivery.

Available methods for AVM geometry reconstruction from neuroimaging data (CT, MRI, cerebral angiography) make it possible to distinguish vessels with an average diameter of at least 0.5 mm. It is insufficient to determine in vivo the detailed geometric AVM structure, which consists of a very large number of intertwined thin vessels whose diameter can reach up to 0.1 mm. Therefore, when modeling AVMs, it becomes necessary to use simplified approaches. The paper considers a one-dimensional model of AVM embolization, which allows us to describe important qualitative patterns of this process. In addition to the difficulty of AVM geometry reconstruction, the mathematical description of blood rheology, arterial and venous vessels walls properties, the forces arising in vessels walls, as well as the forces acting on the vessel from the surrounding tissues, the filtration processes through the vessel wall, chemical reactions, and other ongoing processes are still far away to completion. Despite this, taking into account the most significant and determining parameters of the system allows one to build mathematical models that give a qualitatively correct description of its behavior.

For a more detailed study of embolization in future, it is necessary to consider more accurate models that take into account additional characteristics of the AVM embolization process. This primarily concerns the uneven distribution of the physical characteristics of the AVM (porosity, permeability, and cross-sectional area) along model length. More accurate embolic agent behavior description is also important since some of its varieties are liquids with pronounced non-Newtonian properties. In addition, embolic agent adsorption in blood vessels walls and its properties variation over time during solidification are of interest. Despite the impossibility at present of complete AVM internal structure restoration using neuroimaging data, macroscopic characteristics such as the shape of cross sections and the length of AVM nidus can be restored with sufficient for modeling accuracy. In this regard, the next stage of the study may

A Clinically based AVM geometry and permeabilities

Philips ComboMap system and the Philips ComboWire sensor (sensor diameter is 0.36 mm and its length is 1.85 m) were used in neurosurgical operations to measure blood velocity and pressure inside the cerebral vessels near the pathology. This way the values of pressure and velocity v_b in AVM inlet (artery) were obtained before, during and after the embolization for 10 patients. Further data on geometrical AVM parameters such as length L and cross-section area A as well as inlet artery cross-section area ω are used. This parameters are obtained based on perioperative X-ray tomography. Further three patients K , T and P with the largest number of intraoperative measurements are considered.

From (3) one can see that Buckley-Leverett function is completely determined by the relative phase permeabilities and viscosities of the blood and embolic agent. Known blood viscosity [11] and common embolic agent ONYX18 [7] are:

$$\eta_b \approx 4 \text{ cP}, \quad \eta_e \approx 18 \text{ cP}. \quad (22)$$

To calculate blood relative phase permeability its absolute and phase permeabilities are needed. Absolute permeability K is porous medium (AVM nidus) characteristic. It is assumed to be constant and could be calculated using monitoring data before embolization while embolic agent is not in the AVM and it is completely filled with blood. From Darcy's law we get:

$$K = -\frac{Q_{b_0} \eta_b}{A \frac{\Delta p_0}{L}} \equiv \text{const}, \quad (23)$$

where $Q_{b_0} = v_{b_0} \omega$ is blood volume flow before embolization, v_{b_0} - blood flow velocity in the arterial AVM inlet before embolization, Δp_0 - pressure drop between arterial AVM inlet and venous AVM outlet before embolization. Values Q_{b_0} and Δp_0 are based on monitoring data.

Total AVM embolization was achieved in three considered operations, assuming that in total embolization the AVM nidus is filled by embolic agent completely. Monitoring data includes velocity, pressure and injected embolic agent fraction. Using information on the total amount of embolic agent used for the operation and its fractions in the intermediate time points t_i we can calculate volume average blood concentration $\bar{S}(t_i)$ in the AVM.

Assuming that embolic agent is always distributed uniformly among AVM blood phase permeability K_b at time t_i is given by formula:

$$K_b(\bar{S}_i) = -\frac{Q_b(\bar{S}_i) \eta_b}{A \frac{\Delta p(\bar{S}_i)}{L}}, \quad (24)$$

where $\bar{S}_i = \bar{S}(t_i)$, $Q_b(\bar{S}_i)$ is blood flow and $\Delta p(\bar{S}_i)$ is pressure drop between artery and vein during embolization. Values $Q_b(\bar{S}_i) = v_b(\bar{S}_i) \omega$ and $\Delta p(\bar{S}_i)$ are also calculated using monitoring data.

A Clinically based AVM geometry and permeabilities

Philips ComboMap system and the Philips ComboWire sensor (sensor diameter is 0.36 mm and its length is 1.85 m) were used in neurosurgical operations to measure blood velocity and pressure inside the cerebral vessels near the pathology. This way the values of pressure and velocity v_b in AVM inlet (artery) were obtained before, during and after the embolization for 10 patients. Further data on geometrical AVM parameters such as length L and cross-section area A as well as inlet artery cross-section area ω are used. This parameters are obtained based on perioperative X-ray tomography. Further three patients K , T and P with the largest number of intraoperative measurements are considered.

From (4) one can see that Buckley-Leverett function is completely determined by the relative phase permeabilities and viscosities of the blood and embolic agent. Known blood viscosity [11] and common embolic agent ONYX18 [7] are:

$$\eta_b \approx 4 \text{ cP}, \quad \eta_e \approx 18 \text{ cP}. \quad (23)$$

To calculate blood relative phase permeability its absolute and phase permeabilities are needed. Absolute permeability K is porous medium (AVM nidus) characteristic. It is assumed to be constant and could be calculated using monitoring data before embolization while embolic agent is not in the AVM and it is completely filled with blood. From Darcy's law we get:

$$K = -\frac{Q_{b_0} \eta_b}{A \frac{\Delta p_0}{L}} \equiv \text{const}, \quad (24)$$

where $Q_{b_0} = v_{b_0} \omega$ is blood volume flow before embolization, v_{b_0} - blood flow velocity in the arterial AVM inlet before embolization, Δp_0 - pressure drop between arterial AVM inlet and venous AVM outlet before embolization. Values Q_{b_0} and Δp_0 are based on monitoring data.

Total AVM embolization was achieved in three considered operations, assuming that in total embolization the AVM nidus is filled by embolic agent completely. Monitoring data includes velocity, pressure and injected embolic agent fraction. Using information on the total amount of embolic agent used for the operation and its fractions in the intermediate time points t_i we can calculate volume average blood concentration $\bar{S}(t_i)$ in the AVM.

Assuming that embolic agent is always distributed uniformly among AVM blood phase permeability K_b at time t_i is given by formula:

$$K_b(\bar{S}_i) = -\frac{Q_b(\bar{S}_i) \eta_b}{A \frac{\Delta p(\bar{S}_i)}{L}}, \quad (25)$$

where $\bar{S}_i = \bar{S}(t_i)$, $Q_b(\bar{S}_i)$ is blood flow and $\Delta p(\bar{S}_i)$ is pressure drop between artery and vein during embolization. Values $Q_b(\bar{S}_i) = v_b(\bar{S}_i) \omega$ and $\Delta p(\bar{S}_i)$ are also calculated using monitoring data.

AVM embolization optimal scenario

Patient	$L [10^{-2} \cdot m]$	$A [10^{-4} \cdot m^2]$	$\omega [10^{-6} \cdot m^2]$	$K [m^2]$
K	2.4	4.5	4.5	$8.2 \cdot 10^{-11}$
T	2.2	2.0	4.5	$6.7 \cdot 10^{-8}$
P	3.0	3.1	3.1	$3.7 \cdot 10^{-10}$

Table 1: AVM parameters

Then blood relative phase permeability $k_b(\bar{S}_i)$ is the following:

$$k_b(\bar{S}_i) = \frac{K_b(\bar{S}_i)}{K} = \frac{v_b(\bar{S}_i) \Delta p_0}{v_{b_0} \Delta p(\bar{S}_i)}, \quad (25)$$

Formula (25) shows that $k_b(\bar{S}_i)$ is independent of blood and embolic agent viscosity and AVM geometric properties.

Figure 8 shows monitoring data on arterial velocity and pressure [47] during embolization for patients K , T and P . Pressure increases and velocity decreases during embolization. Venous pressure is assumed to decrease linearly during embolization from 40 mmHg to 7 mmHg [48].

AVM geometrical parameters for three considered patients were determined based on itraoperative X-ray angiography and perioperative X-ray tomography and are presented in Tab.1.

Absolute permeability values used for further calculations are presented in Tab.1. Patient T with permeability two orders greater the pressure drop at the AVM inlet and outlet is less than the drop for patients K and P. Currently available clinical measurements for 10 patients with cerebral AVM allow us to assume that absolute permeability is within $10^{-8} - 10^{-12} m^2$ and for seven patients is of order 10^{-10} . As far as authors know previously value of absolute permeability in vivo for such inaccessible biological objects as AVM was not presented. This bio-mechanical AVM nidus porous medium characteristic was obtained due to in vivo velocity and pressure measurements.

AVM embolization optimal scenario

Patient	$L [10^{-2} \cdot m]$	$A [10^{-4} \cdot m^2]$	$\omega [10^{-6} \cdot m^2]$	$K [m^2]$
K	2.4	4.5	4.5	$8.2 \cdot 10^{-11}$
T	2.2	2.0	4.5	$6.7 \cdot 10^{-8}$
P	3.0	3.1	3.1	$3.7 \cdot 10^{-10}$

Table 1: AVM parameters

Then blood relative phase permeability $k_b(\bar{S}_i)$ is the following:

$$k_b(\bar{S}_i) = \frac{K_b(\bar{S}_i)}{K} = \frac{v_b(\bar{S}_i) \Delta p_0}{v_{b_0} \Delta p(\bar{S}_i)}, \quad (26)$$

Formula (26) shows that $k_b(\bar{S}_i)$ is independent of blood and embolic agent viscosity and AVM geometric properties.

Figure 8 shows monitoring data on arterial velocity and pressure [47] during embolization for patients K , T and P . Pressure increases and velocity decreases during embolization. Venous pressure is assumed to decrease linearly during embolization from 40 mmHg to 7 mmHg [48].

AVM geometrical parameters for three considered patients were determined based on itraoperative X-ray angiography and perioperative X-ray tomography and are presented in Tab.1.

Absolute permeability values used for further calculations are presented in Tab.1. Patient T with permeability two orders greater the pressure drop at the AVM inlet and outlet is less than the drop for patients K and P. Currently available clinical measurements for 10 patients with cerebral AVM allow us to assume that absolute permeability is within $10^{-8} - 10^{-12} m^2$ and for seven patients is of order 10^{-10} . As far as authors know previously value of absolute permeability in vivo for such inaccessible biological objects as AVM was not presented. This bio-mechanical AVM nidus porous medium characteristic was obtained due to in vivo velocity and pressure measurements.

AVM embolization optimal scenario

Patient	κ	data SE
K	1.54	± 0.026
T	1.99	± 0.036
P	2.49	± 0.018

Table 2: Corey model parameter

B Clinically based Buckley-Leverett function construction

For further calculations Corey model [49] was used for the analytic approximation of blood relative phase permeability:

$$k_b(S) = S^\alpha, \quad \alpha \in \mathbb{R}, \quad \alpha > 1 \quad (26)$$

The embolic agent relative permeability $k_e(S)$ is assumed to be symmetric to blood relative permeability in the following sense:

$$k_e(S) = k_b(1 - S). \quad (27)$$

Thus Buckley-Leverett function is the following:

$$b(S) = \frac{S^\alpha / \eta_b}{S^\alpha / \eta_b + (1 - S)^\alpha / \eta_e}, \quad (28)$$

where coefficient κ was calculated using the least square method for clinical data. This type of Buckley-Leverett function guarantees its non-convexity and fulfillment of condition (4).

Buckley-Leverett function approximations were constructed using clinical data for patients of concern. Table 2 shows the obtained Corey model parameter values and standard error (SE) of data recovery.

On figure 9 markers stay for Buckley-Leverett function values obtained by formulas (3),(25),(27) using clinical data and lines show Buckley-Leverett approximations by formula (28).

References

- [1] Stephen L. Ondra, Henry Troupp, Eugene D. George, and Karen Schwab. The natural history of symptomatic arteriovenous malformations of the brain: a 24-year follow-up assessment. *J. Neurosurg.*, 73(3):387–391, sep 1990. doi: 10.3171/jns.1990.73.3.0387.
- [2] Robert D. Brown, David O. Wiebers, Glenn Forbes, W. Michael O’Fallon, David G. Piegras, W. Richard Marsh, and Robert J. Maciunas. The natural history of unruptured intracranial arteriovenous malformations. *J. Neurosurg.*, 68(3):352–357, mar 1988. doi: 10.3171/jns.1988.68.3.0352.

AVM embolization optimal scenario

Patient	κ	data SE
K	1.54	± 0.026
T	1.99	± 0.036
P	2.49	± 0.018

Table 2: Corey model parameter

B Clinically based Buckley-Leverett function construction

For further calculations Corey model [49] was used for the analytic approximation of blood relative phase permeability:

$$k_b(S) = S^\alpha, \quad \alpha \in \mathbb{R}, \quad \alpha > 1 \quad (27)$$

The embolic agent relative permeability $k_e(S)$ is assumed to be symmetric to blood relative permeability in the following sense:

$$k_e(S) = k_b(1 - S). \quad (28)$$

Thus Buckley-Leverett function is the following:

$$b(S) = \frac{S^\alpha / \eta_b}{S^\alpha / \eta_b + (1 - S)^\alpha / \eta_e}, \quad (29)$$

where coefficient κ was calculated using the least square method for clinical data. This type of Buckley-Leverett function guarantees its non-convexity and fulfillment of condition (5).

Buckley-Leverett function approximations were constructed using clinical data for patients of concern. Table 2 shows the obtained Corey model parameter values and standard error (SE) of data recovery.

On figure 9 markers stay for Buckley-Leverett function values obtained by formulas (4),(26),(28) using clinical data and lines show Buckley-Leverett approximations by formula (29).

References

- [1] Stephen L. Ondra, Henry Troupp, Eugene D. George, and Karen Schwab. The natural history of symptomatic arteriovenous malformations of the brain: a 24-year follow-up assessment. *J. Neurosurg.*, 73(3):387–391, sep 1990. doi: 10.3171/jns.1990.73.3.0387.
- [2] Robert D. Brown, David O. Wiebers, Glenn Forbes, W. Michael O’Fallon, David G. Piegras, W. Richard Marsh, and Robert J. Maciunas. The natural history of unruptured intracranial arteriovenous malformations. *J. Neurosurg.*, 68(3):352–357, mar 1988. doi: 10.3171/jns.1988.68.3.0352.

Appendix B

Associate Editor Comments to Author (Dr Oliver Jensen):

Please ensure that remaining grammatical errors are fixed. The referee has identified a handful of them but there are others that need to be addressed also.

Reviewer comments to Author:

Reviewer: 1

Comments to the Author(s)

The authors have addressed the majority of my comments and so I am happy to recommend publication provided the language is improved. I cannot list all potential typos/errors but here are some examples/suggestions:

using "than" instead of "then"

Just above equation (1) "Denote local.... " -> "We denote the local... "

Just above equation (2) " Under this assumptions..."

Missing articles throughout

for example - just above equation (7) ", determined by operation strategy" "determined by the chosen clinicail/embolisation strategy"

Dear Dr Oliver Jensen,

We are very grateful for your work and comments given. We did our best to fix the remaining grammatical errors and typos. We also ask you to pass our thanks to the referee, whose informative comments have significantly improved our article.

Kind regards, authors